# PROCEEDINGS A

mechanical engineering, mathematical modelling

snaking, localization, shell buckling

**Author for correspondence:**
Giles W. Hunt
e-mail: g.w.hunt@bath.ac.uk

# Maxwell tipping points: the hidden mechanics of an axially compressed cylindrical shell

G. W. Hunt[1,2], R. M. J. Groh[2] and T. J. Dodwell[3,4]

[1]Department of Mechanical Engineering, University of Bath, Bath, UK
[2]Faculty of Engineering, University of Bristol, Bristol, UK
[3]College of Engineering, Mathematics and Physical Sciences, University of Exeter, Exeter, UK
[4]The Alan Turing Institute, London NW1 2DB, UK

RMJG, 0000-0001-5031-7493; TJD, 0000-0003-0408-200X

Numerical results for the axially compressed cylindrical shell demonstrate the post-buckling response snaking in both the applied load and corresponding end-shortening. Fluctuations in load, associated with progressive axial formation of circumferential rings of dimples, are well known. Snaking in end-shortening, describing the evolution from a single dimple into the first complete ring of dimples, is a recent discovery. To uncover the mechanics behind these different phenomena, simple finite degree-of-freedom cellular models are introduced, based on hierarchical arrangements of simple unit cells with snapback characteristics. The analyses indicate two fundamentally different variants to this new form of snaking. Each cell has its own Maxwell displacement, which are either separated or overlap. In the presence of energetic background disturbance, the differences between these two situations can be crucial. If the Maxwell displacements of individual cells are separated, then buckling is likely to occur sequentially, with the system able to settle into different localized states in turn. Yet if Maxwell displacements overlap, then a global buckling pattern triggers immediately as a dynamic domino effect. We use the term *Maxwell tipping point* to identify the point of switching between these two behaviours.

# 1. Introduction

Nonlinear instabilities in elastic structures such as thin plates and shells have been the focus of considerable effort over the years, both before and after Koiter's ground-breaking thesis published in Dutch [1] at the end of the Second World War. Koiter's work lay largely dormant until translated into English at the start of the 1960s, following which, fuelled by new design challenges from aerospace engineering in particular, the burst of activity was such that it might have seemed that, to all intents and purposes, the story was complete. However, the fact that important phenomena—localization, spatial chaos and snaking for example—remained undiscovered for another two decades, pays testament to the subtleties that such systems possess. Different types of post-buckled solutions can coexist, and the least energy, practically relevant, localized forms have, at times, been masked by more readily obtained periodic counterparts. In the course of the on-going development, it is arguably true that no single structural problem has played a more central role than the buckling of the long and thin cylindrical shell under axial compression; its capacity for symmetry-driven mode interaction [1,2], together with its notorious sensitivity to small imperfections and disturbances [3], mark it out for special attention. Here, with the buckling modes defined from the linearized problem as harmonic both axially and circumferentially, it is not surprising that traditional approaches have tended to be Fourier-based and come up with periodic solutions, once they have entered the post-buckling regime.

This paper uses the cylinder problem to draw attention to a behaviour that has received little or no attention in the past, that of sideways or transverse snaking. Its counterpart, longitudinal snaking, was identified in the late 1990s as a generic sequence of discretely occurring localized buckles occurring along the length of a long structure [4,5]. In the dynamical phase-space sense, this behaviour was described by a series of homoclinic orbits [6], leading in the long run to a heteroclinic connection between the unbuckled state and the periodic buckle pattern that exists at the classical Maxwell load [5,7]; it was found in a number of different problems [5], perhaps most notably the 1D Swift–Hohenberg equation [4]. For the cylinder, the initial homoclinic solution of the sequence was shown to take the form of a single circumferential ring of axially localized buckles appearing in a central region away from the ends, with the more advanced homoclinics describing the addition of further rings [8].

It has, however, recently been revealed that the single row of buckles of the initial homoclinic shape is itself the end result of a more localized snaking process, starting with a single dimple and progressing around the circumference (orthogonal to the loading direction) with extra dimples being added in turn. While the longitudinal snaking sequence has the load rising and falling under growing end-displacement and is centred about the classical *Maxwell load*, this second sequence is marked by the end-displacement oscillating back and forth under falling load, and is centred about its dual, the *Maxwell displacement*. One notable difference between these two snaking sequences is that, while the former typically can appear in one dimension, the latter clearly requires two.

The fact that a single localized dimple can initiate the buckling process in a cylinder is a fairly recent observation. It was suggested initially by Horák, Lord, & Peletier [9] and later demonstrated in numerical solutions to the von Kármán–Donnell equations by Kreilos & Schneider [10]. Both these analyses considered bifurcation from a linear fundamental equilibrium path comprising pure axial squash and uniform dilation, a circumstance that would be hard to realize in a physical experiment because of boundary constraints at the ends of the shell. In numerical investigations such hidden mechanics have largely remained undiscovered. Explicit finite-element formulations suffer from the same challenges as experimental tests, i.e. unstable equilibria cannot be traced, while implicit schemes, based on pseudo arc-length path-following methods, have previously modelled the cylinder as multiple ODEs defined axially, coupled by periodic circumferential modes [5]. The latter approach naturally eliminates the localized single dimple as an admissible solution, and only recently Groh & Pirrera [11] have demonstrated the behaviour in a high-fidelity finite-element formulation using path-following continuation methods. A set of results from this analysis are shown here in figure 1, and demonstrate that

**3**

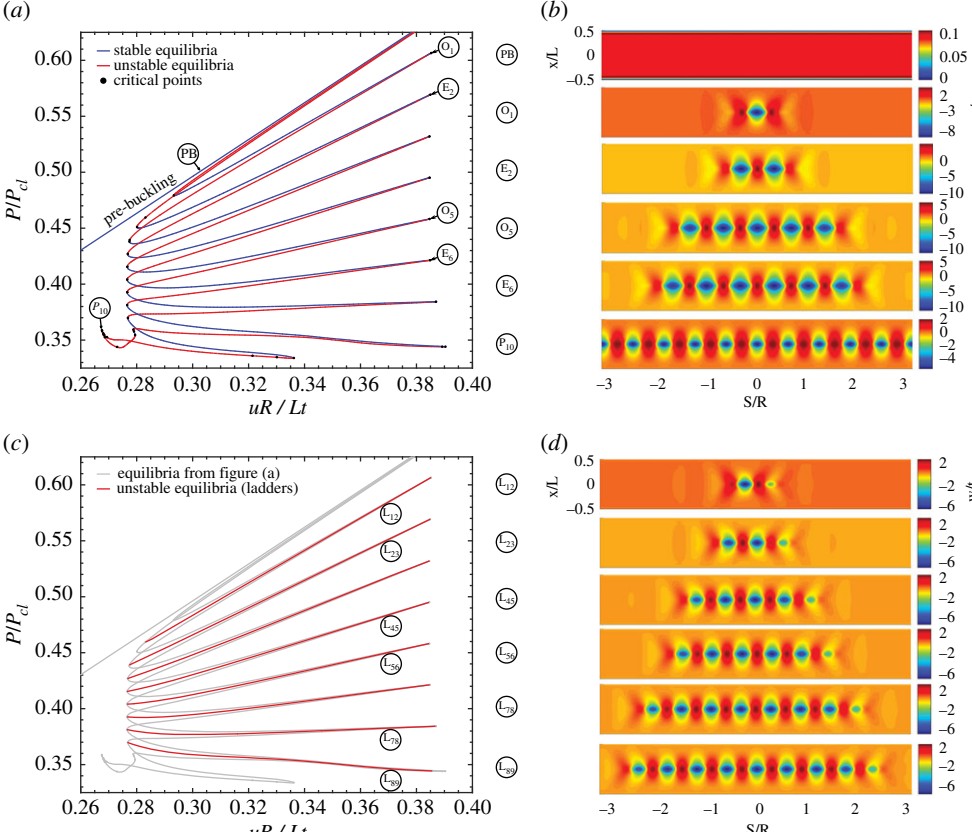

**Figure 1.** Equilibrium paths and pertinent mode shapes of snaking and laddering in the cylinder's post-buckling regime. (*a*) Equilibrium path of odd and even post-buckling solutions growing sequentially around the cylinder circumference through series of destabilizations and restabilizations. (*b*) Deformation mode shapes for odd-numbered and even-numbered localized solutions culminating in a single row of ten buckles. (*c*) Equilibrium path of the unstable ladders (or rungs) connecting the odd-numbered and even-numbered dimple waveforms. (*d*) Pertinent deformation mode shapes on the ladders illustrating their symmetry-breaking effect. (Online version in colour.)

although the fundamental and post-buckling equilibrium paths lie close to one another on a load *vs* end-shortening plot, they remain disconnected, meaning that direct bifurcation from pre- to post-buckling is absent. This effect is a result of choosing practically relevant clamped boundary conditions, which means that uniform dilation is violated close to the ends.

As with longitudinal snaking [5], it is instructive to explore the character of the phenomenon using uncoupled or weakly coupled low-dimensional cells. In earlier studies, such cells were placed in series with each cell carrying the same load [12]. In this particular arrangement, individual cells buckle-through sequentially with the pattern formation occurring in the direction of the applied load. As the buckling sequence progresses, the applied load rises and falls with the *Maxwell load* serving as an organizing centre for the load fluctuation.

Transverse snaking, i.e. a pattern formation orthogonal to the applied load, can be demonstrated with the same kind of cellular system, but this time with individual cells placed in parallel. It is instructive to assume that each cell has a tri-linear response made up of three different positive stiffnesses. When uncoupled, in the well-known analogy to Ohm's law, resultant stiffnesses can readily be computed by adding component stiffnesses, either directly (for the parallel system) or according to a reciprocal relation (for the series system). In the case

of transverse snaking, the *Maxwell displacement* plays the role as an organizing centre for the displacement fluctuation.

As shown in figure 1, the equilibrium states of the single dimple and subsequent snaking scenario are mostly unstable under rigid loading. Even though the cylinder can be perturbed onto regions where, say, a single dimple is statically stable, as observed by Esslinger [13], such states are nevertheless hard to find experimentally in a robust manner. So why then are they of practical significance? The answer lies in the nature of the triggering buckling mechanism. Work by Thompson & Sieber [14] focuses on the issue by analytically 'probing' with a conceptual point load, and measuring the work required to push the cylinder through to a single dimple. The energy barrier resisting this process can be visualized on a plot of load against its corresponding probe displacement, and the magnitude of the energy barrier provides a measure of the associated *shock sensitivity*; i.e. the ease with which the cylinder can transition out of the pre-buckling state via the single dimple by means of static or dynamic perturbations. Once an initial input of energy has managed to overcome this barrier, the Maxwell displacement marks the point at which the same energy is subsequently dynamically released to promote the next buckle in the sequence. This suggests that the instability sequence could progress as a domino effect, with the released energy each time being just sufficient to trigger the next buckle. In such a way, uncovering the full range of equilibria is essential to expose the otherwise hidden mechanics of cylinder buckling.

We show herein, that a two-dimensional grid made up of destabilizing/restabilizing cells, based on two degree-of-freedom arches placed in series and in parallel, is enough to illustrate the full range of cylinder behaviour. It is found that whether the complete instability sequence happens spontaneously as a cascade of localized buckles, or needs a steady increase in applied end-shortening, depends on the degree of coupling between adjacent cells. Weak or zero coupling between adjacent cells in a row means that buckling in a single cell is likely to cause instantaneous buckling of an entire row. Alternatively, strong coupling between cells acts to stabilize the response, such that at each stage extra energy is required to promote the next local buckling event. These characteristics, which might at first seem counterintuitive, are examined in detail in §4a(ii) and §4c. The point, i.e. the level of external loading, at which the system switches between these two fundamentally different paradigms, is what we term the *Maxwell tipping point*.

The paper proceeds as follows. We start with recent results along the lines of Groh & Pirrera [11], reproduced from a recent review of unstable buckling [15]. The resulting plot of load against end-shortening demonstrates the snaking behaviour of interest, with the load falling while end-shortening is fluctuating throughout the pattern formation. Next, by postulating a grid of four uncoupled cells placed in series and in parallel, a typical overall behaviour is described according to the resulting computed stiffnesses. A four-cell model comprising two degree-of-freedom arches is then presented, highlighting the difference between weakly and strongly coupled systems and describing the associated Maxwell tipping point. The paper closes by revisiting the cylinder problem in the light of these new developments.

## 2. Localization in a thin-walled axially compressed cylindrical shell

A thin-walled isotropic cylindrical shell of thickness $t = 0.247$ mm, radius $R = 100$ mm and length $L = 160.9$ mm, which is loaded in uniform axial compression via displacement control, is considered. The cylinder is modelled as linearly elastic and isotropic with Young's modulus $E = 5.56$ GPa and Poisson's ratio $v = 0.3$, representing Yamaki's [16] longest cylinder (Batdorf parameter $Z = L^2\sqrt{1 - v^2}/Rt = 1000$). To replicate a typical experimental set-up, the cylinder is rigidly clamped at both ends with uniform axial compression imposed at one end only. Hence, at one end of the cylinder all six degrees of freedom are constrained, whereas at the other end, five degrees of freedom are constrained with axial end-shortening a free parameter varied during the analysis.

The cylinder is modelled using isoparametric, geometrically nonlinear finite elements based on a total Lagrangian formulation. The finite elements used are so-called 'degenerated shell

elements' [17] based on first-order shear deformation theory assumptions [18]. The full cylinder is discretized into 193 axial and 480 circumferential nodes that are assembled into 25-noded spectral finite elements defined by Payette & Reddy [19] and the large rotation parametrization of shell normals described by Bathe [20]. The circumferential domain, $s$, is described by $s/R \in [-\pi, \pi]$ and the axial domain, $x$, by $x/L \in [-0.5, 0.5]$. In figure 1 below, blue segments denote stable equilibria, red segments denote unstable equilibria, and black dots are critical points (limit points and branching points). For additional details on the modelling set-up and the numerical continuation algorithms used, see refs [11,21].

Figure 1$a$ shows an equilibrium manifold of applied end-shortening $u$ (normalized by the cylinder length, radius and thickness) $vs$ the ensuing reaction load $P$ (normalized by the classical buckling load $P_{cl} = 2\pi Et^2/\sqrt{3(1-\nu^2)}$). The corresponding deformation mode shapes at the points specified in figure 1$a$ are shown in figure 1$b$ on an 'unrolled' cylinder. Running diagonally in blue in figure 1$a$ is the trivial pre-buckling path of uniform axial squashing and circumferential dilation; see mode shape (PB) in figure 1$b$. Running almost coincident to the pre-buckling path are two unstable solutions; one corresponding to a single dimple in the cylinder wall and the other to a double-dimple solution. Both these equilibrium paths undergo a snaking sequence whereby additional buckles are added to the left and right of the original pattern (thereby maintaining symmetry) through a series of de- and restabilizations, i.e. the single-dimple solution grows in the odd-integer sequence $1, 3, 5, \ldots, 9$, while the double-dimple solution grows in an even-integer sequence $2, 4, 6, \ldots, 10$. The two snaking sequences are intertwined and ultimately coalesce at the same point ($P_{10}$ in figure 1$a$) where the snaking sequence terminates in a single row of ten buckles. As shown by Groh & Pirrera [11], point $P_{10}$ is a branching point where the two snaking sequences connect to an extended stable segment of the ten-buckle waveform.

As previously mentioned in §1, interesting features of this snaking sequence are that (i) the pattern formation occurs orthogonal to the loading direction, and (ii) the snaking sequence fluctuates within a region of applied end-shortening ($0.276 < uR/Lt < 0.391$), while the reaction force $P$ is falling with each additional buckle in the waveform. This is significantly different from earlier work on the cylinder [8], where it was shown that a snaking sequence causing pattern formation along the length of the cylinder, i.e. the sequential formation of complete rings of buckles in the direction of the applied load, leads to an increase in end-shortening while the reaction force fluctuates. The latter behaviour has been recognized for many years as a generic sequence typical of many mechanical systems [5,7], but its dual manifestation, which has end-displacement fluctuating while the load drops, is explored here in detail for the first time; we shall refer to it as *transverse snaking*.

However, as is usual in homoclinic snaking [7], the two intertwined odd- and even-numbered snaking sequences—which in this case preserve left–right and up–down symmetry in the mode shapes—are connected by ladders (or rungs) that break the left–right symmetry group. As shown in figure 1$c$, the ladders connect the odd- and even-numbered snaking sequences via unstable equilibrium paths connecting a branching point on one snake (close to $uR/Lt \approx 0.28$) to a branching point on the other snake (close to $uR/Lt \approx 0.39$). The deformation modes in figure 1$d$ clearly show the symmetry-breaking effect of the ladders, whereby a dimple is only added to one side of the current pattern to connect odd- and even-numbered waveforms.

As the ladders represent unstable equilibria, they can never manifest themselves as stable, observable states in practice. However, unstable equilibria—such as the ladders and the unstable portions of the two intertwined snaking sequences—form energy barriers between stable equilibria. For example, to transition from the pre-buckling state to the post-buckling state with a single dimple, say for $uR/Lt = 0.34$ in figure 1$a$, the system has to traverse the energy barrier associated with the unstable equilibrium path running parallel to the stable pre-buckling path. Similarly, once the cylinder is stable in the post-buckling state with one dimple, it can transition into other stable states by traversing other energy barriers. For example, to transition from one stable dimple to two stable dimples, the cylinder would have to overcome the energy barrier provided by ladder $L_{12}$ in figure 1$c$. Alternatively, it could transition from one stable dimple to three stable dimples by traversing the unstable portion of the odd-numbered snake in

figure 1*a*, termed $S_{13}$. Indeed, if sufficient disturbance energy is input into the system, the cylinder might transition immediately into higher-order buckling modes, but in this case, multiple energy barriers and valleys would need to be traversed.

The above scenario adds significantly to the known complexities surrounding cylinder buckling. To aid understanding, in the following sections we demonstrate that the behaviours of snaking, laddering and associated Maxwell displacements can be replicated using a hierarchical arrangement of snapping cells. Even though these systems are discrete analogues to the continuum cylinder, they retain the fundamental mechanics and flavour of the problem; namely, a snaking sequence with pattern formation transverse to the direction of applied loading, with reaction force falling and end-shortening oscillating. We start by outlining some simple underlying characteristics of such systems, which can be obtained directly from well-known stiffness relations.

## 3. Stiffness computations for tri-linear cells

### (a) Elements in series

For linear springs placed in series such that they all carry the same load, it is well known that a resultant stiffness is readily obtained by reciprocal addition of individual stiffnesses. If such springs are given a bi-linear or tri-linear characteristic, then depending on the status of the individual springs a number of different resultant stiffnesses are available. When individual cells switch stiffness one-by-one, as in cellular buckling [5], it is then possible to trace a multi-linear overall response on a load *vs* end-shortening plot, based on computations of resultant stiffnesses alone.

Figure 2, reproduced from an earlier publication [5], demonstrates this behaviour. Figure 2*a* depicts four cells configured in series such that they all carry the same load, but are otherwise uncoupled. Each cell has a tri-linear response comprising an initial (infinite) pre-buckling stiffness, a negative post-buckling stiffness $-k_1$, and a final positive finite (restabilized) stiffness $k_2$, as seen in figure 2*b*. The response on a plot of load $P$ against total end-shortening $\Delta$ shown in figure 2*c*, at any stage, then depends on just two possibilities. If $m$ individual elements are at positive stiffness $k_2$ with the remainder at the pre-buckled infinite stiffness, then the response is on a rising path with resultant stiffness, $R_m$, where

$$\frac{1}{R_m} = \frac{m}{k_2}. \tag{3.1}$$

If, on the other hand one spring is at stiffness $-k_1$, while $m$ others are at positive stiffness $k_2$, the resultant stiffness $F_m$ is given by

$$\frac{1}{F_m} = \frac{m}{k_2} - \frac{1}{k_1} = \frac{1}{R_m} - \frac{1}{k_1}, \tag{3.2}$$

which can be positive or negative. Although in the absence of coupling the system is anti-integrable and the order of buckling cannot be specified, the overall response depends simply on the cells buckling one-by-one in turn. With increasing $m$, the pattern of equilibrium paths shown as light lines in figure 2*c* is uniquely sketched out. Under dead load, all of the falling (negative stiffness) portions of this path would be unstable, while under rigid load (controlled end-shortening) each portion would be stable when the overall stiffness is negative but unstable once it turns positive and 'snapback' becomes possible. The heavy line gives the response under controlled end-shortening under the assumption that energy jumps are initiated at the corresponding Maxwell displacement, as described in figure 3*b*.

Under controlled end-shortening, in the absence of coupling the 'snapback' events of figure 2*c* occur in unspecified order along the length. With rigid pre-buckling there is initially no snapback and no Maxwell displacement. As more cells buckle, the triggering load at the Maxwell displacement drops and the response leans over further to the right. $A_1$ and $A_2$ of figure 3*b* give

**7**

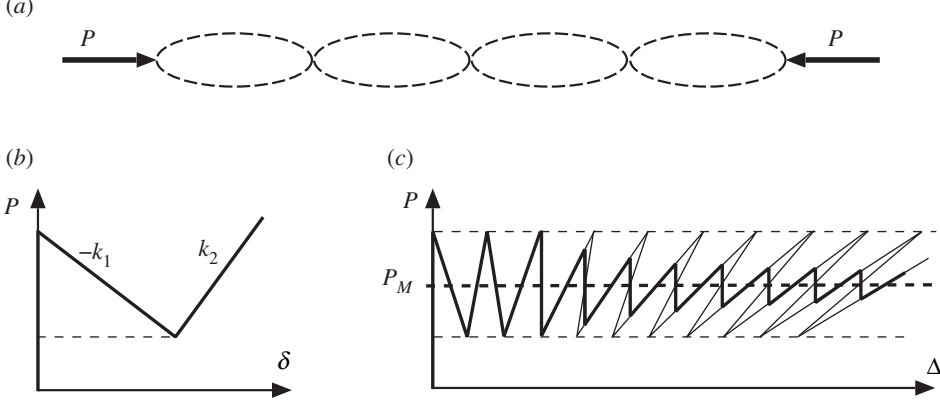

**Figure 2.** Snaking solution for multiple cells in series. (*a*) Configuration of four cells. (*b*) Assumed tri-linear load-shortening response of a typical cell. (*c*) Combined response for ten cells or more. The light curve denotes the as-computed snaking behaviour, $P_M$ is the Maxwell load, and the bold curve is the likely behaviour as individual cells buckle in turn at their Maxwell displacement (figure 3). (Reproduced from [5] with slight change in notation.).

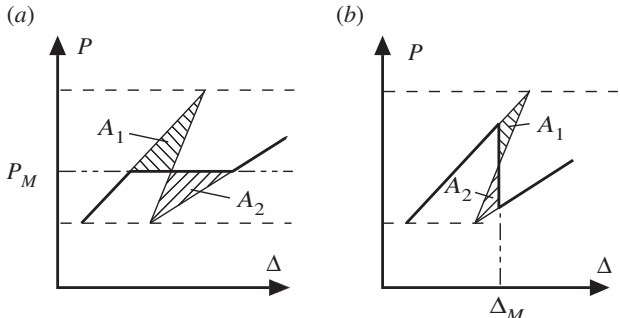

**Figure 3.** Typical segment of figure 2*c*. (*a*) Controlled load: $A_1 = A_2$ for the Maxwell load $P_M$. (*b*) Controlled end-shortening: $A_1 = A_2$ for the Maxwell displacement $\Delta_M$. (Reproduced from [5] with slight change in notation.)

respectively the energy hump to be overcome in escaping from the minimum in the initial state, and the energy released as the system settles into the minimum of the buckled state, and are readily computable [5]; $A_1 = A_2$ defines the Maxwell displacement $\Delta_M$, where the energies in the two minima are equal.

## (b) Elements in parallel

The next objective is to repeat the description for similar elements in parallel under controlled end-displacement $\Delta$, before moving on to two-dimensional arrays as seen in figure 4*a*. Parallel sets of cells are assumed to displace by the same $\Delta$, but are otherwise uncoupled. The chosen load-deflection response for a single cell is shown in figure 4*b*, here given a tri-linear characteristic, similar to that shown in figure 2*b* except with a finite initial stiffness to allow for snapback. With the combined stiffness for different combinations of $k_1$, $k_2$ and $k_3$ being readily computable by summing individual stiffnesses, the response of two cells in parallel must be as shown in figure 4*c*. As with cells in series, a piecewise linear form of snaking is produced, except that it now is arranged vertically around the Maxwell displacement rather than horizontally about the Maxwell load.

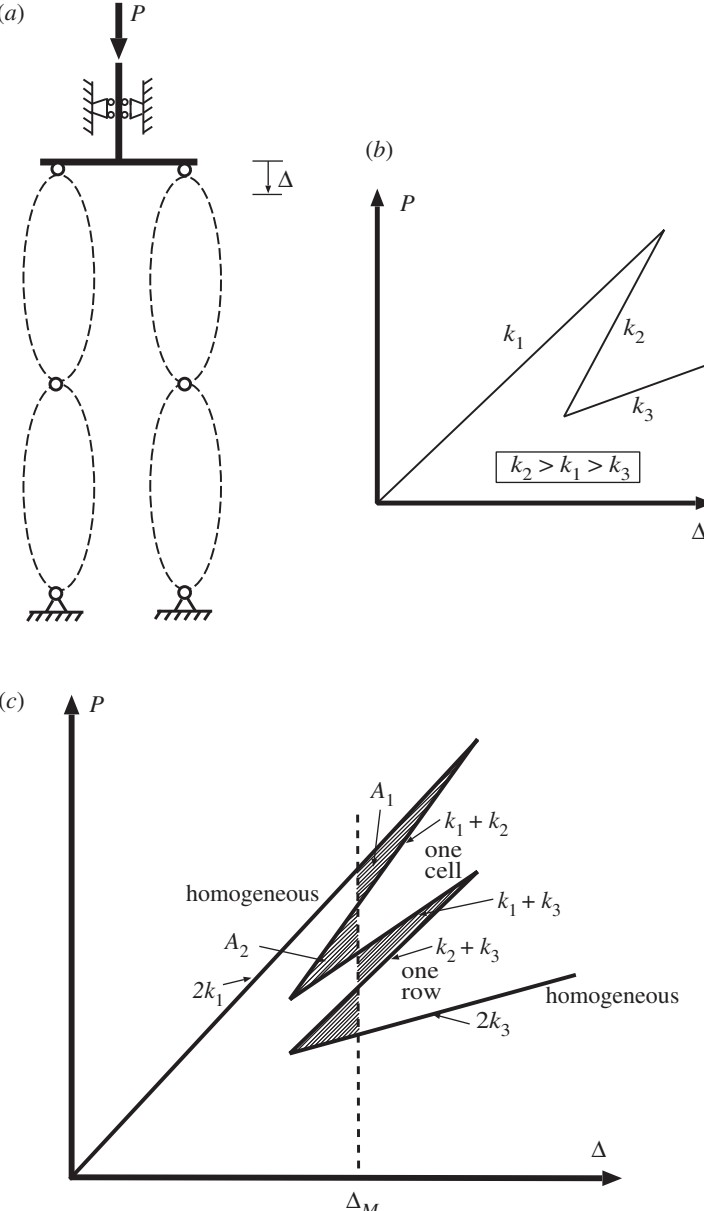

**Figure 4.** (*a*) Two-dimensional array of anti-integrable elements similar to those in figure 2*a*. (*b*) Load–displacement characteristic of a single cell. (*c*) Combined load to corresponding deflection response for two cells in parallel. Maxwell displacement $\Delta_M$ defined when $A_1 = A_2$.

For two cells in parallel, at the Maxwell displacement $\Delta_M$ energy levels at all four stable states—fundamental (stiffness $2k_1$), buckled (stiffness $2k_3$), and two non-homogeneous half-buckled (stiffness $k_1 + k_3$)—are the same. The two non-homogeneous unstable configurations (stiffnesses $k_1 + k_2$ and $k_2 + k_3$), represent the energy humps separating these stable states. It is immediately apparent that half the energy input is required to trigger just one of the instabilities, or indeed two instabilities one after the other, than it is to trigger both simultaneously. The Maxwell displacement $\Delta_M$, where $A_1 = A_2$, is as shown. For $\Delta > \Delta_M$, one or the other of the stable non-homogeneous states is the most likely outcome under a sudden disturbance or shock.

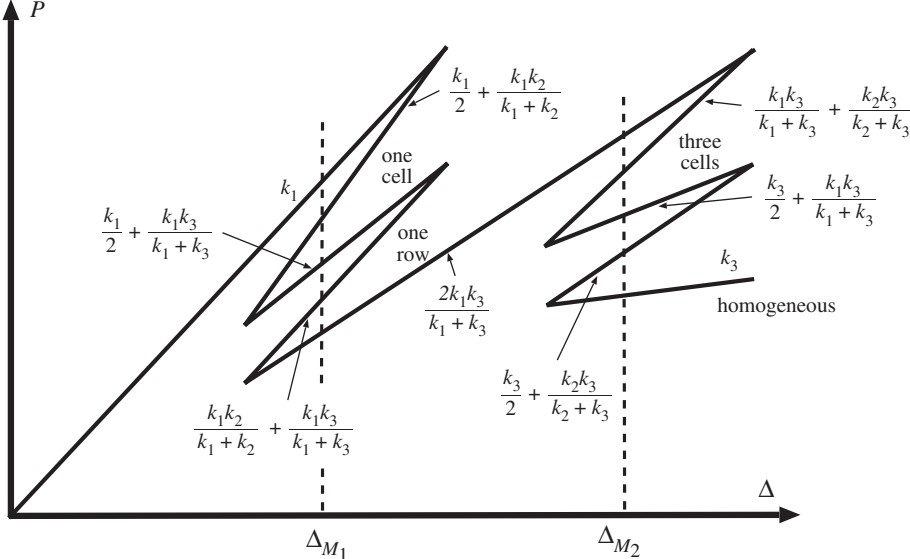

**Figure 5.** Equilibrium paths of the 2-by-2 system of figure 4a.

Assuming that the system remains conservative so that energy is not irreversibly lost (to heat, friction, etc.), once the first instability is complete, the same level of disturbing energy remains available to trigger the second instability. It should be noted however, that under conditions of controlled end-displacement and in the absence of coupling terms between the elements, transfer of energy between elements may not be possible, so the second instability is by no means guaranteed.

### (c) Series and parallel

Moving on to the 2-by-2 array of elements seen in figure 4a, by analogy with Ohm's law for resistors/capacitors, the final stiffness for any given combination may be computed either by summing stiffnesses for parallel elements, or via a reciprocal law for elements in series. The paths of figure 5 are the result. Under controlled end-displacement, one row of elements first completes the snaking process around its related Maxwell displacement $\Delta_{M_1}$, as in figure 4c. The system then restabilizes under positive stiffness until it reaches a second Maxwell displacement $\Delta_{M_2}$, whereupon it follows a second snaking sequence in the previously unbuckled row.

## 4. Snapback arches

We next look for a physical system made up of cells with similar stiffness characteristics. It is relatively easy to come up with a rigid-link and spring model that has a bi-linear response [22], but a tri-linear system is harder to devise. Nonetheless, although simple stiffness calculations are now out of the question, the tied arch set in series with a linear spring of figure 6a, behaves in a similar way. The total potential energy can be written

$$V = U - PL\Delta = \frac{1}{2}kL^2 \left( 2\sqrt{1 - Q^2} - 2\sqrt{1 - h^2} \right)^2 + \frac{1}{2}k_2 L^2 (\Delta - h + Q)^2 - PL\Delta,$$

where $U = (1/2)kL^2(2\sqrt{1 - Q^2} - 2\sqrt{1 - h^2})^2 + (1/2)k_2L^2(\Delta - h + Q)^2$ is the strain energy stored in springs $k$ and $k_2$, and the final term is the work done by the dead load $P$. By differentiating $V$ with respect to the two degrees-of-freedom $Q$ and $\Delta$, two equilibrium equations are produced. These solve to give the load–deflection characteristic of figure 6b, which has an initially stable

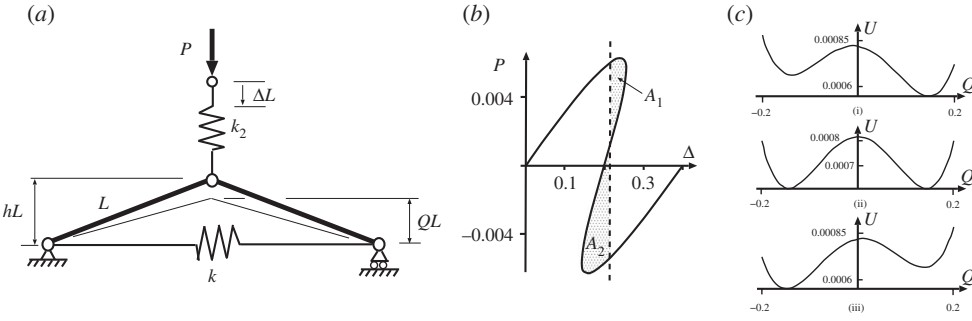

**Figure 6.** (*a*) Tied arch with spring in series, shown in the unstressed state. (*b*) Snapback response shown for $h = 0.2$, $k = 1$ and $k_2 = 0.04$. $A_1$ and $A_2$ represent, respectively, the energy barrier and energy available for release in moving from the upper equilibrium position to the lower position. (*c*) Energy landscape at different values of $\Delta$. (i) $A_1 > A_2$ at $\Delta = 0.19 < \Delta_M$. (ii) $A_1 = A_2$ at Maxwell displacement $\Delta_M = 0.2$. (iii) $A_1 < A_2$ at $\Delta = 0.21 > \Delta_M$.

equilibrium path passing over a maximum limit point and into a region where it goes back on itself, and then through a minimum limit point into a new region of positive stiffness. Like the tri-linear system of figure 4, it has three distinct positive-stiffness regimes—two that are stable and one that is unstable even under conditions of controlled displacement $\Delta$. At the fixed displacement $\Delta$ marked by the dashed line, $A_1$ represents the energy hump to be overcome via the unstable equilibrium before the system can move to the second stable equilibrium state, and $A_2$ is the energy released once this hump has been traversed. The condition $A_1 = A_2$ again marks the *Maxwell displacement*, where the two stable states have equal energy.

## (a) Elements in parallel

### (i) Uncoupled system

For a row of two cells that are uncoupled apart from a frictionless connecting hinge, as seen in figure 7a, the potential function becomes

$$
V_{\text{uncoupled}} = \frac{1}{2}kL^2 \left(2\sqrt{1 - Q_1^2} - 2\sqrt{1 - h^2}\right)^2 + \frac{1}{2}kL^2 \left(2\sqrt{1 - Q_2^2} - 2\sqrt{1 - h^2}\right)^2
$$
$$
+ \frac{1}{2}k_2L^2(\Delta - h + Q_1)^2 + \frac{1}{2}k_2L^2(\Delta - h + Q_2)^2 - PL\Delta. \tag{4.1}
$$

Differentiation with respect to the three degrees of freedom $Q_1$, $Q_2$ and $\Delta$ gives three solvable equilibrium equations and lead to the response of figure 7b. The smooth path mirrors that of figure 6b with equal displacements in each cell, while bifurcations mark a breaking of this symmetry and provide an additional non-homogeneous solution $Q_1 \neq Q_2$.

As in figure 4, there are now two potential instabilities, one in each cell, each taking place at the Maxwell displacement $\Delta_M$. The inherent symmetry suggests that all four areas $A_i$ for $i = 1..4$ are equal. Again it is immediately apparent from the areas involved that the required level of disturbing energy required to initiate buckling in a single cell, or in one cell followed by the other, is half that needed for both cells to buckle through simultaneously.

Figure 8 shows the load/displacement response for four uncoupled cells in parallel, constrained to act together as in figure 7. The added complexity that comes with a relatively modest increase in the number of degrees of freedom from three to five is immediately apparent. The initial loading path, distinguished here by ④ and shown in black, has all four cells deflecting by the same amount at all stages. Starting from the origin, the path passes over a limit point and encounters a multiple bifurcation point $B$, from which emerge a number of equilibrium paths; these are distinguished by having either one, two or three cells deflecting by the same amount

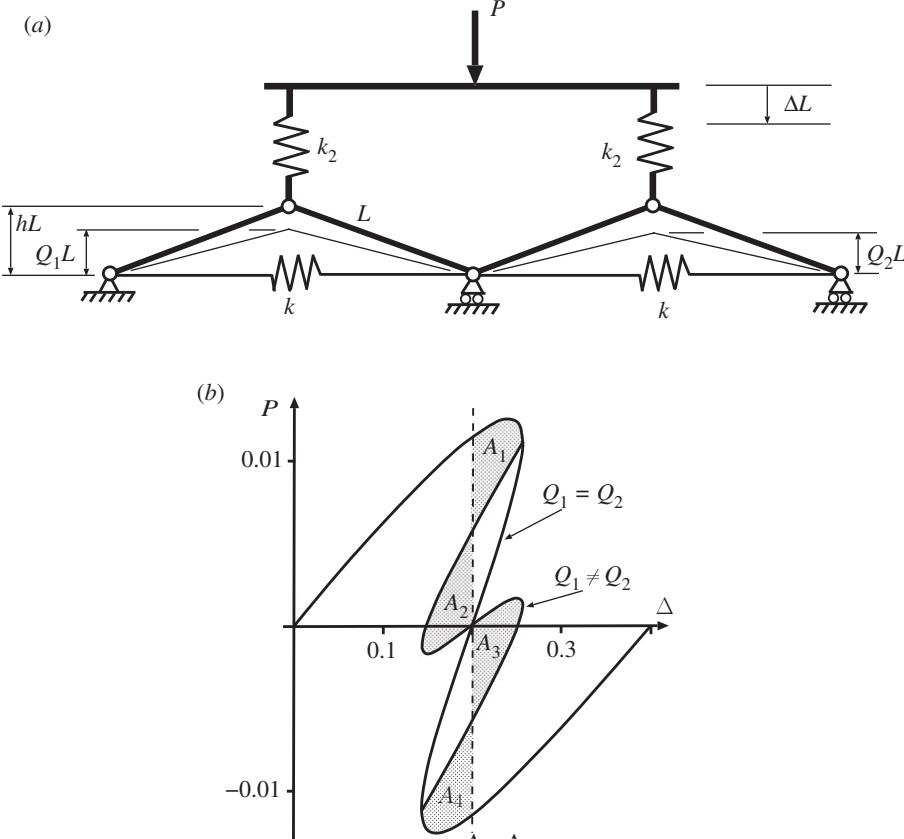

**Figure 7.** Uncoupled cells in parallel. (*a*) Physical system. (*b*) Load–displacement response demonstrating single Maxwell displacement $\Delta_M$ when $A_1 = A_2 = A_3 = A_4$. Drawn for $L = 1, h = 0.2, k = 1$ and $k_2 = 0.04$.

and more markedly than the rest, and are respectively labelled ① (in cadet blue), ② (in red) and ③ (in green). As with the cylinder plot of figure 1, ladder solutions, marked as Ⓛ and shown in blue, also exist, transitioning between the numbered solutions and providing energy humps to be overcome during buckling. The extra complexity of this complete picture of the equilibrium paths means that in some of the plots which follow, we only show immediately relevant paths.

### (ii) Coupled systems

Coupling between elements in parallel can be accomplished in a number of ways, most simply by adding a rotational spring to the common pin position as seen in figures 9 and 10. The main effect of this addition is to shift the positions of the two Maxwell displacements relative to one another as the cells buckle in turn. As $\Delta$ moves from $\Delta_{M_1}$ to $\Delta_{M_2}$, the difference between $\Delta_{M_1}$ to $\Delta_{M_2}$ can be either positive, as seen in figure 9 or negative as in figure 10. The difference is significant in that, if it is assumed that the first instability occurs at the Maxwell displacement ($\Delta = \Delta_{M_1}$) or shortly thereafter, for the former set-up a finite increase in $\Delta$ is required to get from $\Delta_{M_1}$ to $\Delta_{M_2}$, while for the latter case, once the initial instability has taken place at or beyond $\Delta_{M_1}$, the system immediately has enough stored energy to trigger the second instability.

Both systems have the total potential energy for the uncoupled system of figure 7, $V_{\text{uncoupled}}$ of equation (4.1), with an additional term for the strain energy of the coupling spring, namely,

$$U_c = \tfrac{1}{2} K \left( 2 \arcsin h - \arcsin Q_1 - \arcsin Q_2 \right)^2 \tag{4.2}$$

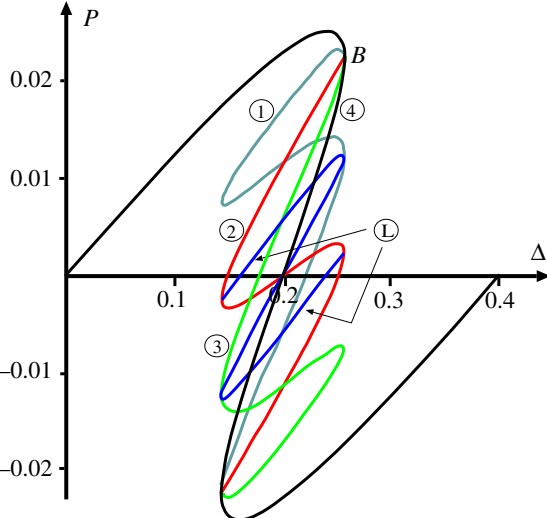

**Figure 8.** Response of four uncoupled cells, set in parallel and constrained to deflect the same amount $\Delta$, as in figure 7. (Online version in colour.)

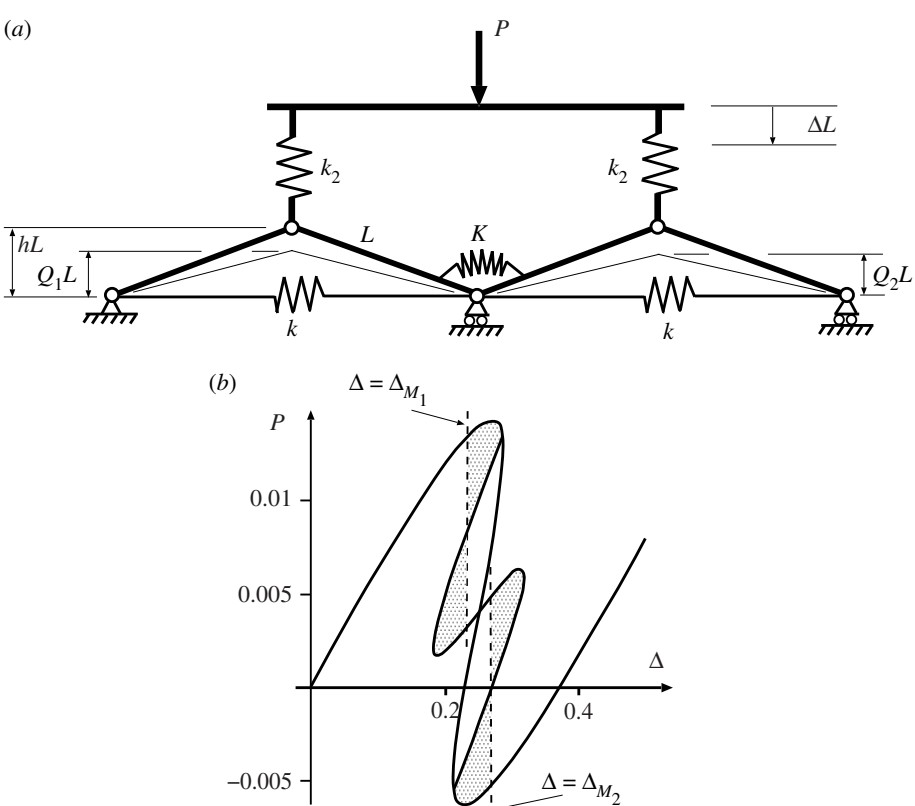

**Figure 9.** Sequential Maxwell displacements. Under controlled $\Delta$, if instability occurs at or soon after the first Maxwell displacement $\Delta_{M_1}$, the system restabilizes in a localized state. Here the coupling spring $K$ is unstressed when $\arcsin Q_1 + \arcsin Q_2 = 2 \arcsin h$. Shows (a) system configuration and (b) load-deflection response for $L = 1$, $h = 0.2$, $k = 1$, $k_2 = 0.04$ and $K = 0.005$.

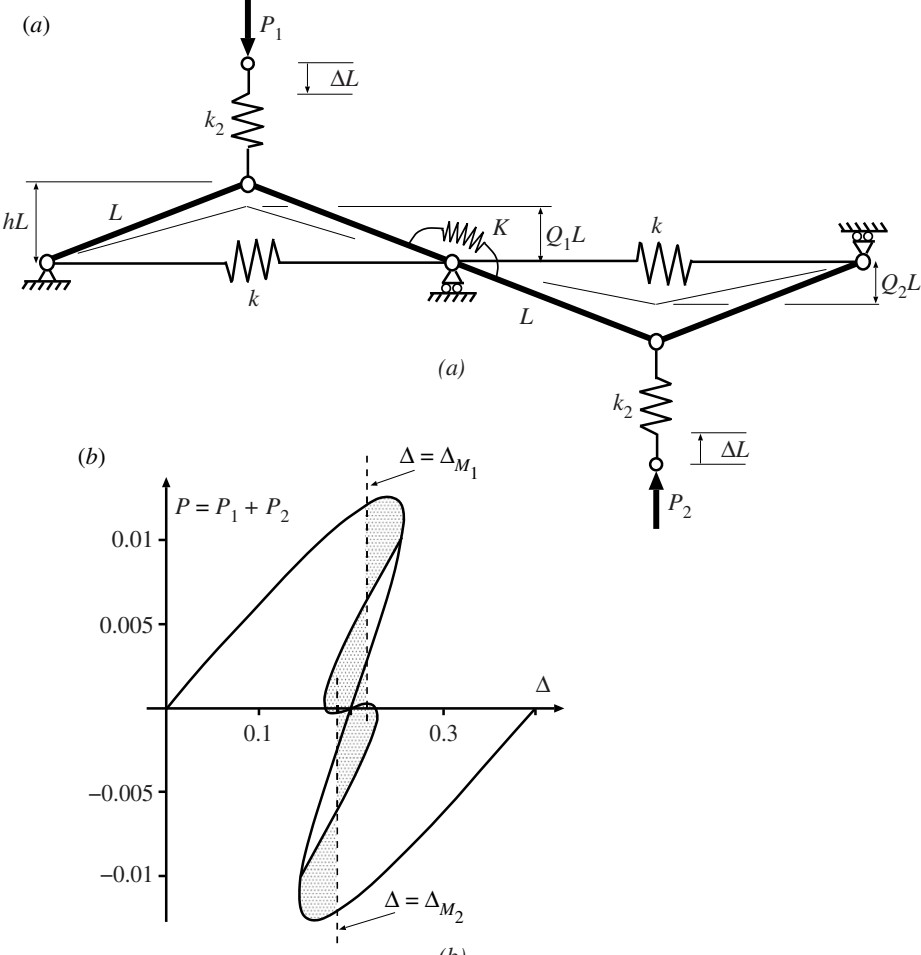

**Figure 10.** Overlapping Maxwell displacements. Under controlled $\Delta$, if the first instability occurs at or soon after the first Maxwell displacement $\Delta_{M_1}$, it can immediately trigger the second instability, as the energy released in the first is sufficient to overcome the next energy hump. Here the coupling spring $K$ is unstressed when $\arcsin Q_1 = \arcsin Q_2$. Shows (a) system configuration and (b) load-deflection response for $L = 1, h = 0.2, k = 1, k_2 = 0.04$ and $K = 0.005$.

for the system of figure 9 and

$$U_c = \tfrac{1}{2}K(\arcsin Q_1 - \arcsin Q_2)^2 \tag{4.3}$$

for that of figure 10. The difference in the responses comes from this energy term as the system passes from the initial homogeneous state, through the non-homogeneous localized transition and into the second homogeneous state. Equation (4.2) shows that $U_c$ increases with either $Q_1$ or $Q_2$, whereas equation (4.3) has $U_c$ depending on the difference, $Q_1 - Q_2$; so $Q_1 = Q_2$ means that the coupling spring stores zero energy for figure 10 but not for figure 9. In comparison with the uncoupled case of figure 8, the Maxwell displacement for the first instability, $\Delta_{M_1}$ is therefore delayed in both situations, reflecting the extra energy needed to overcome a larger energy hump. However, while for figure 9 the second instability, $\Delta_{M_2}$, is further delayed, reflecting the further increase in $U_c$ necessarily found in the final homogeneous state, for figure 10 the opposite is the case. Now, the energy level of $U_c$ drops back to zero in the final state where $Q_1 = Q_2$, and so at $\Delta_{M_1}$ the energy level of this final minimum must lie below that of the first. This means that $\Delta_{M_2}$ has already been exceeded by the time $\Delta$ gets to $\Delta_{M_1}$, and the Maxwell displacements overlap as shown.

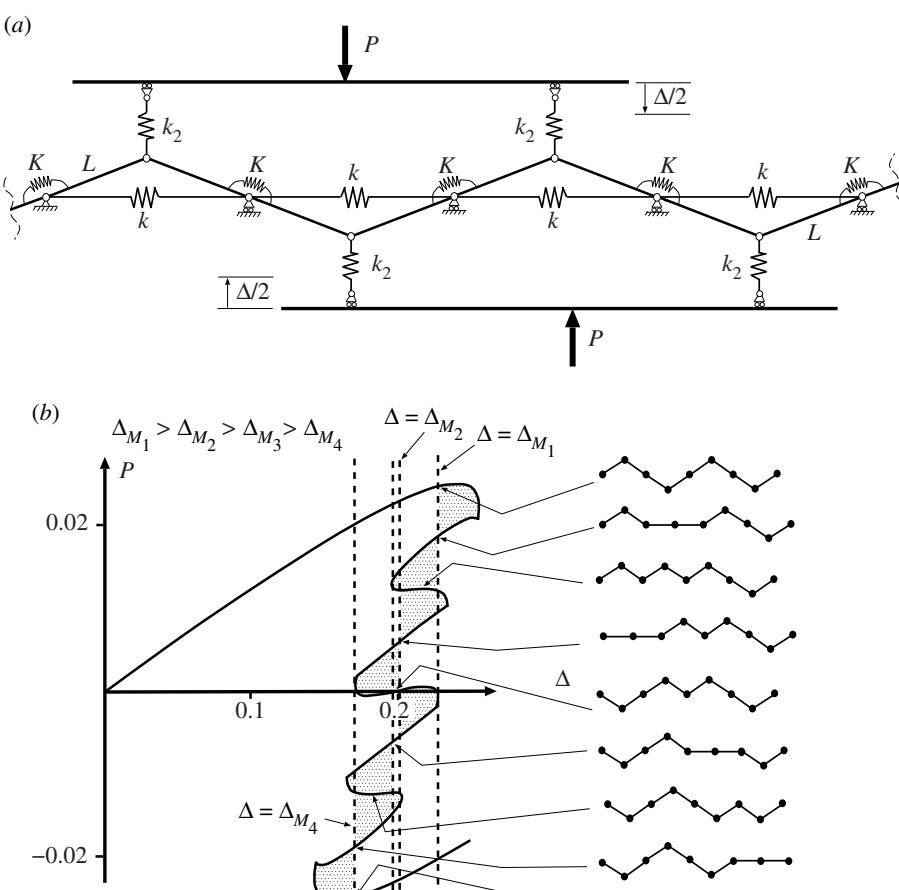

**Figure 11.** (*a*) Four coupled cells in parallel. (*b*) Load–displacement response, showing overlapping Maxwell displacements. Plotted for $L = 1, h = 0.2, k = 1, k_2 = 0.04$ and $K = 0.005$.

When the system of figure 10 is extended to four cells, as shown in figure 11, an interesting pattern starts to emerge. Here we take a little 'modelling licence', in that the individual cells are positioned in-line, yet the end cells are assumed to be linked by a coupling spring, as though they are arranged in a loop; in eliminating boundary effects this reflects the circumferential geometry of the cylinder, but also avoids complex three-dimensional geometries associated with out-of-plane curvature, which are not needed for this phenomenological description. The corresponding potential function is given by

$$V = \frac{1}{2}kL^2 \sum_{i=1}^{4}\left(2\sqrt{1-Q_i^2} - 2\sqrt{1-h^2}\right)^2 + \frac{1}{2}k_2L^2\sum_{i=1}^{4}(\Delta - h + Q_i)^2$$

$$+ \frac{1}{2}K\left(\sum_{i=1}^{3}(\arcsin Q_{i+1} - \arcsin Q_i)^2 + (\arcsin Q_4 - \arcsin Q_1)^2\right). \quad (4.4)$$

It is instructive to compare the response of figure 11*b* with the uncoupled paths of figure 8; because the tangle of equilibrium paths becomes even more complex in figure 11*b*, only those of direct relevance to the expected buckling sequence are shown. The second deflected shape on the right of figure 11*b* is found on the path marked ① in figure 8; the active cell is second from the

left in the row, but of course it could have have been any one of the four. The associated Maxwell displacement $\Delta_{M_1}$—by the mechanism described above for the system of figure 10—is greater than its uncoupled counterpart (at $\Delta = 0.2$), i.e. significant strain energy is being stored in the coupling springs at each end of the active cell, thereby raising the energy hump and pushing the Maxwell displacement forward. The second instability of the sequence, however, taking place in a cell adjacent to the first (here to its left), has a reduced energy hump in comparison with the first. The coupling spring at the left is storing energy, but the one at the other end of the cell is releasing it as they approach the same $Q_i$ value. This results in a second Maxwell displacement, $\Delta_{M_2}$ (associated with the path shown as ② in figure 8) lying much closer to its uncoupled counterpart at $\Delta = 0.2$ than $\Delta_{M_1}$. The third Maxwell displacement $\Delta_{M_3}$ (associated with the path shown as ③ in figure 8) similarly has one spring storing energy while the other releases it, and so again lies close to $\Delta = 0.2$, while the fourth releases energy in both springs, so has $\Delta_{M_4}$ significantly less than $\Delta = 0.2$.

One can image a similar effect to figures 10 and 11 for the cylinder, as there is likely to be more localized stretching energy involved at the interface between a single dimple and the adjoining cylindrical section, than between two adjacent dimples in a periodic sequence. The circumferential interface between two dimples is effectively a bending-dominated fold, whereas the interface at the end of the series involves stretching energy as the doubly curved dimple transitions into the cylindrical shape. This also implies that the largest energy barrier to be overcome in the cylinder is that of initiating a single dimple (two transition regions from doubly curved dimple to cylinder shape). Additional energy barriers via ladders, see figure 1c, only initiate one new transition region and therefore correspond to a smaller energy barrier. This mirrors the particular arrangement and effect of the coupling springs depicted in figure 11a, and we would therefore expect the Maxwell behaviour of the cylinder to behave as depicted in figure 11b.

## (b) Elements in series

Figure 12 shows two cells in series. The system is similar to a recent re-entrant model used to demonstrate changes in phase between auxetic and non-auxetic behaviour, with the addition of a spring in series to lean the equilibrium path over and allow for snapback. The response of figure 12 under controlled displacement $\Delta$ takes the typical form of longitudinal snaking, as described in [5] and reproduced here in figures 2 and 3. Two Maxwell displacements exist, $\Delta_{M_1}$ and $\Delta_{M_2}$, indicating that the two cells will buckle in turn.

## (c) Series and parallel

If two systems as shown in figure 12 are placed in parallel, the bottom two cells are obliged to displace by the same amplitude. To release this unwanted constraint, a second in-line spring $k_2$ can be placed between the lower cells and the hard boundary, giving the system of figure 13.

This configuration carries extra freedom, in that, with all four cells being able to deflect by different amounts, the central part may move upwards or downwards. An extra degree of freedom $\Gamma$ is introduced to allow for this. The issue resonates with cylinder buckling, in that a row of dimples as seen in figure 1 necessarily forms away from the ends where hard boundary conditions prevail.

The potential function for the system of figure 13 can be written

$$V = \frac{1}{2}kL^2 \sum_{i=1}^{4} \left( 2\sqrt{1 - Q_i^2} - 2\sqrt{1 - h^2} \right)^2 + \frac{1}{2}k_2L^2 \left( \sum_{i=1}^{2}(\Delta - \Gamma - h + Q_i)^2 + \sum_{i=3}^{4}(\Gamma - h + Q_i)^2 \right)$$

$$+ \frac{1}{2}K \left( (2\arcsin h - \arcsin Q_1 - \arcsin Q_2)^2 + (2\arcsin h - \arcsin Q_3 - \arcsin Q_4)^2 \right) - PL\Delta,$$

(4.5)

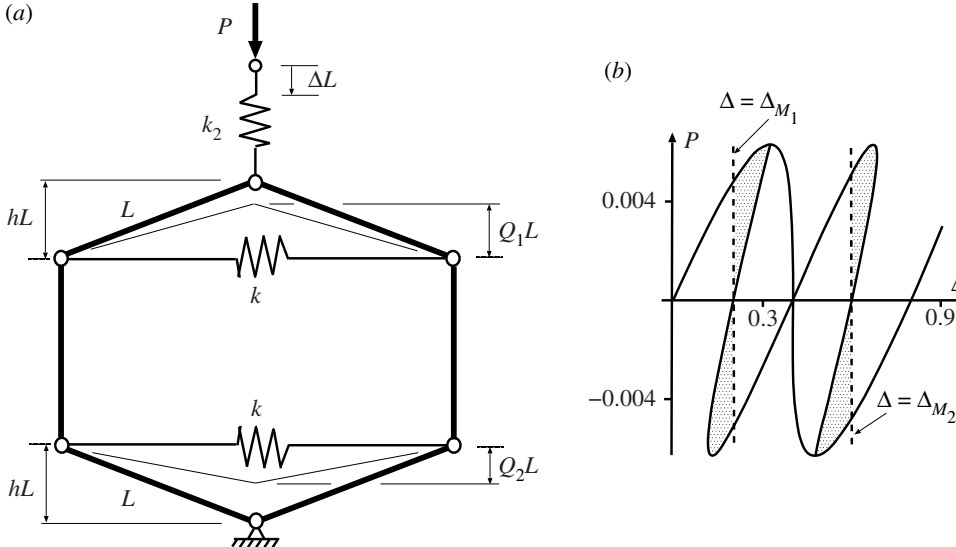

**Figure 12.** Two elements in series. Note similarities with a recent re-entrant auxetic model [23]. Plotted for $L = 1$, $h = 0.2$, $k = 1$ and $k_2 = 0.04$.

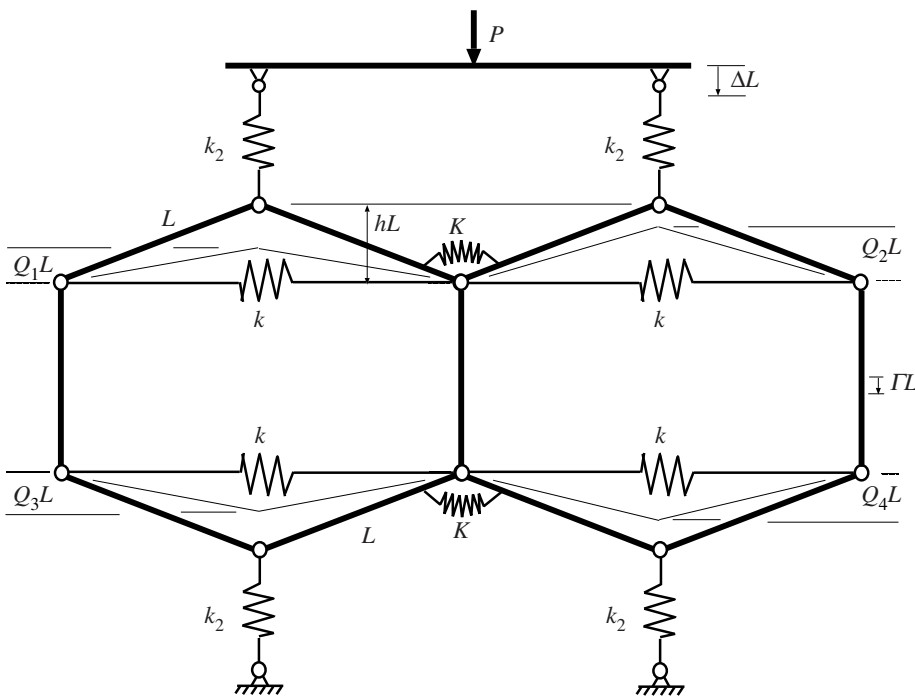

**Figure 13.** Block of four elements. The central part comprising rigid vertical bars moves $\Gamma$ downwards.

and differentiation with respect to the six generalized coordinates, $Q_i$ for $i = 1, \ldots, 4$, $\Delta$ and $\Gamma$, and equating to zero leads to six solvable equilibrium equations.

The uncoupled system of figure 13 with $K = 0$ has the response of figure 14. Here, different colours have been used to distinguish equilibrium paths with differing symmetry properties. The black line is the solution if all four cells deflect by the same amount, and mirrors the response of

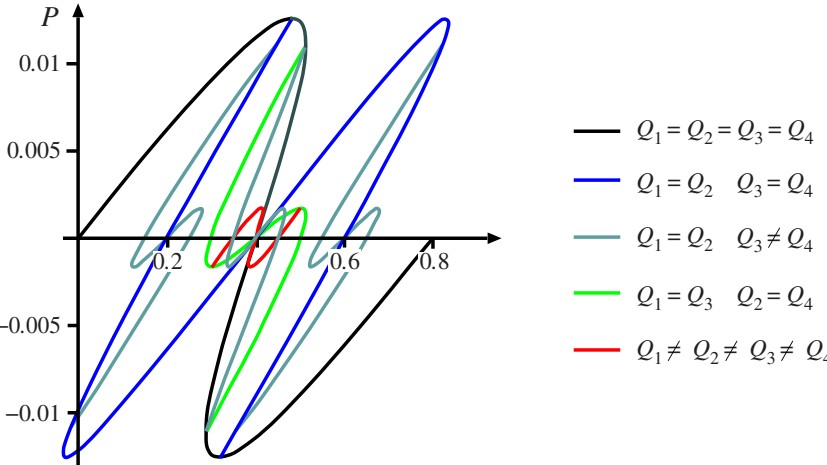

**Figure 14.** Full set of equilibrium paths for the uncoupled four block system of figure 13. Plotted for $h = 0.2$, $k = 1$, $k_2 = 0.04$ and $K = 0$. (Online version in colour.)

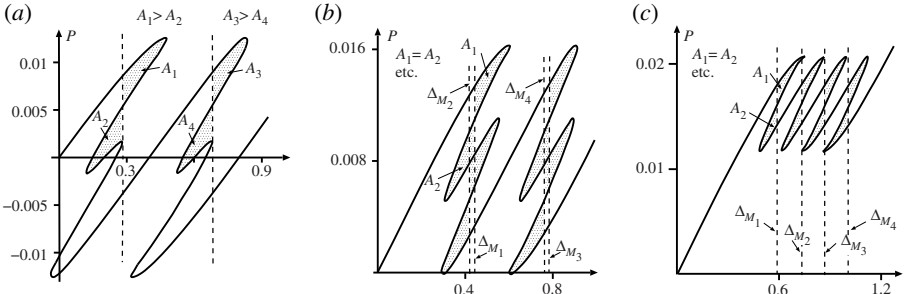

**Figure 15.** Maxwell displacements for the model of figure 13, with only relevant parts of equilibrium paths shown. Rotational springs are unstressed in the zero load position $Q_i = h$. Plotted for $h = 0.2$, $k = 1$, $k_2 = 0.04$. (a) $K = 0$: no Maxwell displacements. (b) $K = 0.01$: overlapping Maxwell displacements per row as in figure 10. (c) $K = 0.02$: sequential Maxwell displacements as in figure 9.

the single cell of figure 6. In addition, the blue line gives the same response as two cells in series, keeping symmetry between the members of each row but allowing localization into either one row or the other. The light green line is the opposite circumstance, maintaining symmetry about the horizontal centreline while allowing localization about the vertical one, and in combination with the black line, mirrors the uncoupled system of figure 7. The dark green lines, bifurcating from either the black or blue lines, break symmetry between the cells of one row but maintain it in the other. Finally, the red line represents solutions in which all the cells deflect by different amounts.

Load–deflection responses for this system with three different values of coupling rotational spring $K$ are shown in figure 15. Here, because of the large number of possible equilibrium paths that exist, only the parts relevant to the Maxwell displacement are shown. Figure 15a shows the paths for the uncoupled case ($K = 0$). Now, although the initial restabilized regions lying below areas $A_2$ and $A_4$ do exist, the system would be unlikely to remain there, as it would arrive with enough excess energy (represented by areas $A_1$ and $A_3$) to pass easily over the next energy barrier.

With moderate coupling ($K = 0.01$), as seen in figure 15b the situation is different, although the outcome under moderate disturbance may end up much the same. Now, four different Maxwell

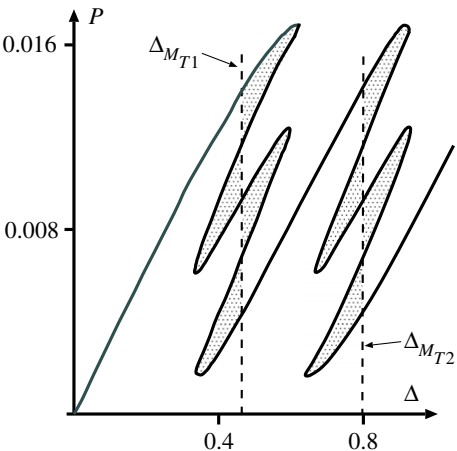

**Figure 16.** Maxwell tipping point for the system of figure 13, at $K = 0.011405$, marking the changeover between overlapping Maxwell displacements, as in figure 15b, and separated Maxwell displacements, as in figure 15c. All the shaded areas are equal in size, indicating that the same amount of energy is required to trigger each instability.

displacements, one for each cell, do exist, and overlap takes place for each row, as seen in figure 10. The complete picture is then akin to that shown in figure 5, with both instabilities in a row occurring at or near the same displacement, but extra end-shortening being required between instability in the first row and the second.

Finally, for strong coupling ($K = 0.02$), the four Maxwell displacements are all separated as shown in figure 15c, with snap-through for both cells of a row having to complete first before snap-through can take place in either cell of the second row.

## (d) Maxwell tipping point

It is clear from figure 15 that a situation exists somewhere in the range $0.01 < K < 0.02$, where the Maxwell displacements for each cell in one row of figure 13 coincide. By trial and error, this circumstance is found to occur at $K = 0.011405$, marking the point where the first instability of one cell in a row releases just enough energy to trigger the instability of the second cell in the same row. The load–deflection response for this special case, which we refer to as the *Maxwell tipping point*, is plotted in figure 16. The end-displacements for two such points, one for each row of the system, are here denoted as $\Delta_{M_{T1}}$ and $\Delta_{M_{T2}}$.

## 5. Cylinder buckling revisited

The phenomena explored in the previous two sections help us understand the snaking and intertwining sequences of figure 1; so we now turn our attention back to the axially compressed cylinder. We start by detailing the two simplest cases of transitioning to the next pattern in a dimpling sequence: (i) the energy barrier associated with traversing from one dimple to two dimples, from two to three, from three to four, etc. via the unstable ladders; and (ii) the energy barrier associated with traversing from one dimple to three dimples, from two to four, from three to five, etc. via the unstable portions of the respective, odd- or even-numbered snake. The ladders are henceforth denoted by $L_{ij}$ and the unstable snaking portions by $S_{ij}$, where each unstable equilibrium path, L or S, represents the energy barrier to traverse from dimple mode $i$ to dimple mode $j$.

As expected intuitively, the energy barrier associated with the ladders, $L_{ij}$, connecting an odd (even) dimple mode to the next even (odd) dimple mode in the sequence is always smaller than

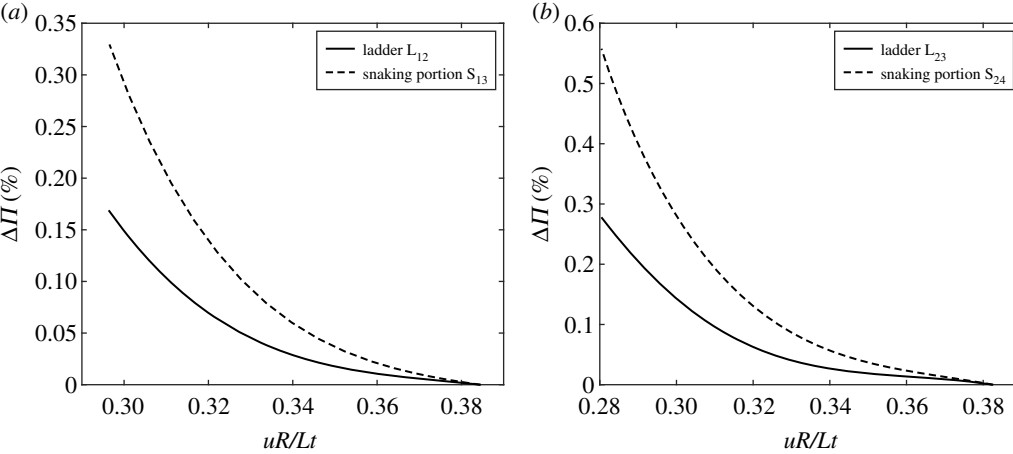

**Figure 17.** Energy barriers ($\Delta \Pi$) of different unstable equilibria, asymmetric ladders and symmetric snakes, with respect to a stable equilibrium path (single dimple and double dimple). (*a*) Energy humps with respect to the stable single-dimple equilibrium path of the unstable ladder ($L_{12}$, one to two dimple connection) and the unstable portion of the snake ($S_{13}$, one to three dimple connection). $\Delta \Pi = 0$ corresponds to the strain energy in the stable single-dimple mode for all levels of $u$. (*b*) Energy humps with respect to the stable two-dimple equilibrium path of the unstable ladder ($L_{23}$, two to three dimple connection) and the unstable portion of the snake ($S_{24}$, two to four dimple connection). $\Delta \Pi = 0$ corresponds to the strain energy in the stable double-dimple mode for all levels of $u$.

the energy barrier associated with the snaking portions, $S_{ij}$, i.e. traversing from an odd (even) dimple mode to the next odd (even) dimple mode. The energy barriers for $L_{12}$, $S_{13}$ and $L_{23}$, $S_{24}$ are compared, respectively, in figures 17*a,b*. In these plots, the *x*-axis denotes the range of compression $uR/Lt$ over which the *i*th mode exists (blue segments in figure 1*a*), and the *y*-axis denotes the percentage difference between the energy in the stable equilibrium (energy well) of the *i*th mode and the energy in the unstable equilibrium (energy hilltop) of the *j*th mode. It is clear that the energy barrier of the ladders, $L_{12}$ and $L_{23}$, is always smaller than the energy barrier of the snaking portions, $S_{13}$ and $S_{24}$, and this trend repeats for all other higher-order modes $L_{ij}$ and $S_{ij}$ (not shown here). In all cases, the energy barrier provided by $L_{ij}$ and $S_{ij}$ with respect to the stable modes *i* is always less than 1% of the strain energy associated with mode *i*, i.e. $\Delta \Pi_{ij} = ((\Pi_j - \Pi_i)/\Pi_i) < 1\%$ for all *u*. This suggests that the stable equilibria along both the odd- and even-numbered snaking sequences are only marginally stable and easily perturbed into other modes by external disturbances, shocks, etc. [14]. Finally, the two sets of curves are seen to intersect and converge towards $uR/Lt \approx 0.39$. This is because in the vicinity of $uR/Lt \approx 0.39$, mode *j* of $L_{ij}$ connects to mode *i* at a branching point, and mode *j* of $S_{ij}$ connects to mode *i* via a limit point (figure 1*c*).

As mentioned in the previous paragraph, the small energy barriers provided by both the unstable ladders and the unstable snaking portions, mean that the cylinder is highly sensitive to external disturbances and imperfections on any of the stable equilibria of the two (odd and even) snaking sequences. Indeed, as shown by Groh & Pirrera [11], the same is true for the pre-buckling path for levels of compression $uR/Lt \gtrsim 0.3$ (or equivalently $u/u_{cl} \gtrsim 0.5$, where the classical buckling end-shortening is given by $u_{cl} = Lt/R \times 1/\sqrt{3(1-\nu^2)}$). As shown in the previous two sections, one way to quantify the high sensitivity to external disturbances and imperfections associated with small energy barriers, and the exceeding likelihood that the cylinder will transition out of marginally stable equilibria, is again the *Maxwell condition* [5]. Here it would mean that the system would be likely to transition out of a marginally stable equilibrium, i.e. a fixed point with vanishingly small energy barriers, if an adjacent stable equilibrium exists with less total potential energy. Hence, for the axially compressed cylinder, we look for the level of end-compression, $u^M$, at which the strain energy of a stable higher-order mode *j* first falls

**Table 1.** Maxwell displacements, $u^M$, for pairs of different dimple buckling modes, e.g. from pre-buckling (0) to one dimple (1), from two dimples (2) to three dimples (3), etc. The classical buckling end-shortening is given by $u_{cl} = Lt/R \times 1/\sqrt{3(1-\nu^2)}$.

| | dimple change | | Maxwell displacement | |
| --- | --- | --- | --- | --- |
| | from mode | to mode | $u^M/u_{cl}$ | $u^M R/Lt$ |
| | 0 | 1 | 0.524 | 0.317 |
| via ladders | 1 | 2 | 0.485 | 0.294 |
| | 2 | 3 | 0.483 | 0.292 |
| | 3 | 4 | 0.482 | 0.292 |
| | 4 | 5 | 0.482 | 0.292 |
| | 5 | 6 | 0.482 | 0.292 |
| | 6 | 7 | 0.482 | 0.292 |
| | 7 | 8 | 0.482 | 0.292 |
| via snakes | 1 | 3 | n.a. | n.a. |
| | 2 | 4 | 0.483 | 0.292 |
| | 3 | 5 | 0.482 | 0.292 |
| | 4 | 6 | 0.482 | 0.292 |
| | 5 | 7 | 0.482 | 0.292 |
| | 6 | 8 | 0.482 | 0.292 |

below the energy of the current marginally stable mode $i$. When this is the case, i.e. the Maxwell displacement $u^M$ is exceeded, the energy barrier provided by ladder $L_{ij}$ or unstable snaking portion $S_{ij}$ is sufficiently small to allow the system to transition spontaneously from the now higher energy well $i$ to the lower energy well $j$.

The results of this energy analysis are summarized in table 1. The first line of the table provides the level of end-shortening $u^M$ at which the total potential of the stable single-dimple buckling mode first falls below the total potential of the pre-buckling state. This occurs at a normalized displacement of $u^M/u_{cl} = 0.524$, which, interestingly, is close to the value $u/u_{cl} = 0.5$ obtained in a linearized buckling analysis of an isotropic cylinder with degraded membrane energy [24,25]. Hence, following the argument of the Maxwell criterion, the cylinder is exceedingly likely to transition out of the stable pre-buckling equilibrium for levels of compression $u/u_{cl} > 0.524$.

Once in the stable single-dimple mode, the cylinder can undergo transitions to higher-order dimple modes either by traversing the energy barrier associated to a ladder or an unstable snaking portion. The results in table 1 show that the sequential Maxwell displacements to higher-order modes, either via the ladders or the unstable snaking portions, tend to a decreasing value of $u^M/u_{cl} = 0.482$. This suggests that the cylinder considered here is in a configuration of overlapping Maxwell displacements (figures 10, 11 and 15b), with the implication that once the first instability is triggered beyond the first Maxwell ($u^M/u_{cl} > 0.524$), enough disturbance energy has been added to the system to trigger all further instabilities. The result is an unstable cascade of dimples forming one by one in turn, finishing in a stable configuration with one fully formed circumferential ring of dimples. Indeed, this dynamic buckling sequence is often described qualitatively in high-speed photography experiments of buckling events in axially compressed cylinders [13]. In summary, our analysis shows that the buckling behaviour of Yamaki's longest cylinder ($Z = 1000$) [16] lies beyond the Maxwell tipping point. This means the cylinder is most likely to stabilize into a fully formed ring of diamonds upon buckling, and this is indeed what is observed in Yamaki's experiments. For other cylinder geometries, i.e. Batdorf parameter $Z$, the buckling behaviour may precede the Maxwell tipping point. Although such a parametric study is

not investigated herein, Yamaki's experiments on other geometries $Z = [50, 100, 200, 500]$ suggest similar behaviour to the case of $Z = 1000$ studied herein [16, p. 236].

We see from table 1 that the largest overlap in Maxwell displacements occurs during the formation of the first dimple. The response of figure 11 for four coupled cells in parallel has a similar trend, and allows us to infer some of the significant characteristics of cylinder buckling. It is clear that the energy barrier to overcome during formation of the first dimple is greater than that for later dimples in the sequence. Analogy with the role of the coupling spring $K$ in figure 11 then suggests that more membrane strain energy is being stored in the cylinder during the first instability than in later instabilities. The first instability involves a transition from folded (dimple) to cylindrical shape at each end of the dimple, whereas the following multi-dimple cases all store energy in just one such transition. Both the cylindrical and folded shapes have zero Gaussian curvature, but transitioning between them is apparently impossible without inducing local membrane stretching. This means that the first instability of the dimpling sequence has an advanced Maxwell displacement in comparison with the others.

# 6. Concluding remarks

We have presented a high-fidelity finite-element analysis of the thin cylindrical shell under controlled axial end-shortening, and demonstrated conclusively that the route to instability starts with a single inwards finite-sized dimple in the surface of the shell, remote from the boundaries. The instability process then progresses with a sequence of further dimples popping inwards in turn, until an entire circumferential row of dimples is created. Under further end-shortening the response then restabilizes, the next stage being a repeat of the sequence into a second row of dimples, adjacent to the first. Most of this behaviour is unstable even under controlled end-shortening, and so has remained largely hidden during experiments, yet it remains absolutely crucial to the buckling process. It identifies, for example, the localized energy hump necessary to trigger buckling into a single dimple, which is then immediately followed by collapse into the full row in a highly unstable domino effect. The related bifurcation sequence is seen to be part of a complex snakes-and-ladders scenario of interacting, odd- and even-numbered localized waves. This process of pattern formation, transverse to the direction of applied loading with the reaction force falling and end-shortening oscillating has, we believe, been discussed in detail here for the first time.

To elucidate the hidden mechanics of this buckling process further, we have introduced a hierarchical arrangement of arch-like mechanisms placed both in series and in parallel. Essentially, each arch corresponds to a cell coupled weakly or strongly to its neighbours in a cellular arrangement. While each individual unit cell features a single limit point both under either force- or displacement-control, the complexity of the response compounds rapidly as the cellular arrangement is scaled up, replicating the response seen in the axially compressed cylinder. First, the snaking sequence in the cellular model corresponds to each cell snapping through individually or alongside other neighbours. Second, the snaking sequence proceeds in rows, i.e. a single cell begins the instability process by snapping through, then cells to the left and right follow, and when a full row is complete the next rows follow by replicating this sequence. Third, by defining a specific degree of coupling between adjacent cells a situation occurs whereby the snap-through of a single cell causes a cascade effect; namely, the energy released by snap-through of a single cell is sufficient to force all other cells in a row to snap through as well. The level of loading at which this occurs is here defined as the *Maxwell tipping point*. Indeed, such behaviour has been observed in experiments of axially compressed cylinders conducted by Esslinger [13]; beyond a specific level of end-shortening, tapping a compressed cylinder with a finger creates a single dimple that then induces a knock-on effect growing into a full ring of dimples. Below a critical threshold of axial compression, this effect does not occur and the single dimple arrests without growing further.

Data accessibility. This article has no additional data.

**Authors' contributions.** All authors contributed to the research, discussed results and implications, and commented on the manuscript at all stages. All authors contributed equally to the idea developement, analysis and writing of this paper.

**Competing interests.** We declare we have no competing interest.

**Funding.** R.M.J.G. is supported by the Royal Academy of Engineering under the Research Fellowship scheme (grant no. RF/201718/17178). T.J.D. is supported by a Turing AI Fellowship (grant no. 2TAFFP/100007). The support of all funders is gratefully acknowledged.

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
