## [Reviewer comments · Proceedings. Mathematical, Physical, and Engineering Sciences]

Review History

RSPA-2020-0273.R0 (Original submission)

Review form: Referee 1

Is the manuscript an original and important contribution to its field?

Excellent

Is the paper of sufficient general interest?

Excellent

Is the overall quality of the paper suitable?

Excellent

Can the paper be shortened without overall detriment to the main message?

Yes

Do you think some of the material would be more appropriate as an electronic appendix?

No

Do you have any ethical concerns with this paper?

No

Recommendation?

Accept with minor revision (please list in comments)

Comments to the Author(s)

The authors have conducted plenty of solid works on the topic of buckling in recent years. The manuscript analyzed the hidden mechanics of this buckling process by introducing a hierarchical arrangement of arch-like mechanisms placed both in series and in parallel. Two fundamentally different variants of snaking are investigated from a new point of view. This work is very interesting and worth of investigation. Only minor revisions are needed to make it ready for publication.

1. Thin-walled structures are generally sensitive to initial geometrical imperfections, which have a significant influence on the post-buckling path. In the proposed framework, how to qualify the stiffness reduction by imperfections is very challenging. Please add some further discussions on this issue.
2. Page 18, line 32, the range of K should be $0.01 < K < 0.02$, rather than $0.1 < K < 0.2$.
3. The reviewer is interested in how is the value of K ($K = 0.011405$) on page 18 determined? Was it derived from the formula or based on the finite element results?
4. In this manuscript, the "Maxwell tipping point" is an important conceptual definition, and the authors give a detailed definition of Maxwell tipping point in Sec. 4 (d). Could you please explain on the role of Maxwell tipping point in the instability of the shell or the physical significance of Maxwell tipping point in the instability of the shell? Because the authors just give the definition based on block of 4 elements shown in Fig. 13., but not based on the buckling behavior of the cylindrical shell.
5. Fig.1 (b) cannot display properly. Please check it.
6. The lines of ① ② ③ in Figure 8 should use different colors to highlight the results. The display of the energy hump display is not very intuitive.

Review form: Referee 2

Is the manuscript an original and important contribution to its field?

Excellent

Is the paper of sufficient general interest?

Excellent

Is the overall quality of the paper suitable?

Excellent

Can the paper be shortened without overall detriment to the main message?

Yes

Do you think some of the material would be more appropriate as an electronic appendix?

No

Do you have any ethical concerns with this paper?

No

Recommendation?

Accept with minor revision (please list in comments)

Comments to the Author(s)

The submission by the authors presents a new insight in the mechanics of the elastic initial post-buckling of an axially-loaded thin-walled cylindrical shell with the clever assistance of spring and

rigid-link models. The insight concerns how the initial row of buckles is produced in a sequence of instabilities that is usually assumed to be triggered instantaneously. However, previously recorded (and indeed well-known) high-speed camera footage has revealed this to be a process involving a sequential buckling phenomenon along the circumference of the cylinder and the present study attributes this to be a "snaking-type", sometimes known as "cellular buckling-type", process. Recent generalized path-following finite element work by one of the co-authors has revealed the highly unstable equilibrium paths that the system contains, although since these are all unstable under rigid loading they are usually missed by physical instrumentation and naked eye observation. In the current work, the Maxwell stability criterion and a new concept, coined as the "Maxwell tipping point" is determined to predict whether a sequential buckling occurs or not.

The submission is very well presented, contains new and tangible mechanical insights, and is recommended for publication in Proceedings A with minor revisions, which are detailed below. (Note that line numbers refer to the numbers on the left hand side of the submitted manuscript).

Pages 3 (line 20) and 9 (line 16): Ohm's law for resistors is an often stated analogy for springs. However, a more exacting analogy is that of capacitors rather than resistors, since capacitors in parallel behave analogously to springs in parallel (effective stiffnesses and capacitances are obtained by summing those of individual elements), while capacitors in series behave analogously to springs in series (effective stiffnesses and capacitances are obtained by summing the reciprocal quantities of individual elements). In resistors, the series and parallel laws are the reverse of the spring stiffness laws. Suggest that this is amended.

Page 4 (line 43): The odd-integer sequence ending in "10" seems a little strange. Presumably "10" is the maximum number of dimples in the cylinder. Suggest that this is rephrased.

Page 5 - Figure 1 (please also see attached file showing marked up page 5):

- 1) In (a) can the black dots in the denoting the critical points be made slightly larger so that they are properly visible?
- 2) In (b), in the PDF that was received, the sequence of images appears to be broken.

Page 5, line 55: Replace "practise" with "practice".

Page 8 - Figure 4(b): There appears to be a misplaced label half-way up the vertical axis. Please check and remove if necessary.

Page 9, lines 7-8: Insert "to" between "(" and "heat, friction, etc.),"

Page 14, final paragraph above Section 4(b), which discusses stretching energies in cylindrical shell dimples was difficult to visualize. Could an additional diagram be included in Figure 11 to assist in visualizing the explanation?

Page 17 (line 31): Surely the range " $0.1 < K < 0.2$ " should be " $0.01 < K < 0.02$ "?

Page 19 (line 14 - Table 1 caption, and line 38): There appear to be an excessive number of closing parentheses in both these expressions.

Page 19 (line 49): Replace "to spontaneously transition" with "to transition spontaneously".

Page 20 (line 46): Replace "to further elucidate the hidden mechanics of this buckling process" with "to elucidate the hidden mechanics of this buckling process further".

Page 21 - Reference 13: Replace "Groh R. M. J." with "R. M. J. Groh", and "Ahmer M. Wadee" with "M. Ahmer Wadee".

Review form: Referee 3

Is the manuscript an original and important contribution to its field?

Excellent

Is the paper of sufficient general interest?

Excellent

Is the overall quality of the paper suitable?

Good

Can the paper be shortened without overall detriment to the main message?

Yes

Do you think some of the material would be more appropriate as an electronic appendix?

No

Do you have any ethical concerns with this paper?

No

Recommendation?

Accept with minor revision (please list in comments)

Comments to the Author(s)

This manuscript focuses on the instantaneous buckling procedure of a thin-walled cylindrical shell under axial compression, from pre-buckling to a single dimple and eventually to a complete ring of dimples along the circumferential direction, which is not easy to be captured experimentally and has drawn little attention. The authors started from the thin cylindrical shell under axial compression using finite element method and then analyzed the deformation mode transition. When instability occurs, a single dimple appears in the shell surface first. In the following process a sequence dimples formed successively. To explore the hidden mechanics, the authors turn to the spring systems coupled with adjacent cells. A two-dimensional cellular model, composed of snap-back arch cells connected in parallel and in series, was proposed to replicate the nonlinear response and illustrate the mechanism behind this phenomenon. The Maxwell displacement plays an important role in the mode selection, which denotes two stable states having equal energy. For some special case, the Maxwell displacement may overlap during multiple modes transformations, which is defined as Maxwell tipping point. This simple model could rationally explain relevant experimental results. The topic of this paper is of interest, which provides a novel insight into the understanding of the complex instability behavior of shell.

Small things:

1. Generally, the Maxwell displacement has a close relationship with the energy of transition modes. It would be better to draw an energy landscape of spring system to provide an overall view and understanding.
2. It seems that Fig. 1(b) is incomplete.
3. Why the initial pre-buckling stiffness of tri-linear cells in series (Fig. 2) is set as infinite, while initial stiffness of cells in parallel (Fig. 4) is finite k_1 ?
4. It is mentioned that the cylinder can undergo transitions from a single dimple to higher-order dimples in two different ways: by traversing the energy barrier associated to a ladder (from odd to even) or an unstable snaking portion (from odd to odd). In experiments which path is observed and why? Can it be controlled experimentally?
5. Earlier literatures have suggested that the Batdorf parameter Z could affect the post-buckling behavior of cylindrical shells. For other smaller or larger Z , can the results be different?
6. "say for $uR/Lt = 0.34$ " in Page 6 may be incorrect according to Fig. 1(a).

Decision letter (RSPA-2020-0273.R0)

20-May-2020

Dear Dr Dodwell,

On behalf of the Editor, I am pleased to inform you that your Manuscript RSPA-2020-0273 entitled "Maxwell tipping points: the hidden mechanics of an axially compressed cylindrical shell" has been accepted for publication subject to minor revisions in Proceedings A. Please find the referees' comments below.

The reviewer(s) have recommended publication, but also suggest some minor revisions to your manuscript. Therefore, I invite you to respond to the reviewer(s)' comments and revise your manuscript. Please note that we have a strict upper limit of 28 pages for each paper. Please endeavour to incorporate any revisions while keeping the paper within journal limits. Please note that page charges are made on all papers longer than 20 pages. If you cannot pay these charges you must reduce your paper to 20 pages before submitting your revision. Your paper has been ESTIMATED to be 22 pages. We cannot proceed with typesetting your paper without your agreement to meet page charges in full should the paper exceed 20 pages when typeset. If you have any questions, please do get in touch.

It is a condition of publication that you submit the revised version of your manuscript within 7 days. If you do not think you will be able to meet this date please let me know in advance of the due date.

To revise your manuscript, log into <https://mc.manuscriptcentral.com/prsa> and enter your Author Centre, where you will find your manuscript title listed under "Manuscripts with Decisions." Under "Actions," click on "Create a Revision." Your manuscript number has been appended to denote a revision.

You will be unable to make your revisions on the originally submitted version of the manuscript. Instead, revise your manuscript and upload a new version through your Author Centre.

IMPORTANT: Your original files are available to you when you upload your revised manuscript. Please delete any redundant files before completing the submission process.

In addition to addressing all of the reviewers' and editor's comments, your revised manuscript **MUST** contain the following sections before the reference list (for any heading that does not apply to your work, please include a comment to this effect):

- Acknowledgements
- Funding statement

See <https://royalsociety.org/journals/authors/author-guidelines/> for further details.

When uploading your revised files, please make sure that you include the following as we cannot proceed without these:

- 1) A text file of the manuscript (doc, txt, rtf or tex), including the references, tables (including captions) and figure captions. Please remove any tracked changes from the text before submission. PDF files are not an accepted format for the "Main Document".
- 2) A separate electronic file of each figure (tif, eps or print-quality pdf preferred). The format should be produced directly from original creation package, or original software format.
- 3) Electronic Supplementary Material (ESM): all supplementary materials accompanying an accepted article will be treated as in their final form. Note that the Royal Society will not edit or typeset supplementary material and it will be hosted as provided. Please ensure that the supplementary material includes the paper details where possible (authors, article title, journal name). Supplementary files will be published alongside the paper on the journal website and posted on the online figshare repository (<https://figshare.com>). The heading and legend provided for each supplementary file during the submission process will be used to create the figshare page, so please ensure these are accurate and informative so that your files can be found in searches. Files on figshare will be made available approximately one week before the accompanying article so that the supplementary material can be attributed a unique DOI. Alternatively you may upload a zip folder containing all source files for your manuscript as described above with a PDF as your "Main Document". This should be the full paper as it appears when compiled from the individual files supplied in the zip folder.

Article Funder

Please ensure you fill in the Article Funder question on page 2 to ensure the correct data is collected for FundRef (<http://www.crossref.org/fundref/>).

Media summary

Please ensure you include a short non-technical summary (up to 100 words) of the key findings/importance of your paper. This will be used for to promote your work and marketing purposes (e.g. press releases). The summary should be prepared using the following guidelines:

- *Write simple English: this is intended for the general public. Please explain any essential technical terms in a short and simple manner.
- *Describe (a) the study (b) its key findings and (c) its implications.
- *State why this work is newsworthy, be concise and do not overstate (true 'breakthroughs' are a rarity).
- *Ensure that you include valid contact details for the lead author (institutional address, email address, telephone number).

Cover images

We welcome submissions of images for possible use on the cover of Proceedings A. Images should be square in dimension and please ensure that you obtain all relevant copyright permissions before submitting the image to us. If you would like to submit an image for consideration please send your image to proceedingsa@royalsociety.org

Once again, thank you for submitting your manuscript to Proceedings A and I look forward to receiving your revision. If you have any questions at all, please do not hesitate to get in touch.

Best wishes
Raminder Shergill
proceedingsa@royalsociety.org
Proceedings A

on behalf of
 Professor Yihui Zhang
 Board Member
 Proceedings A

Reviewer(s)' Comments to Author:

Referee: 1

Comments to the Author(s)

The authors have conducted plenty of solid works on the topic of buckling in recent years. The manuscript analyzed the hidden mechanics of this buckling process by introducing a hierarchical arrangement of arch-like mechanisms placed both in series and in parallel. Two fundamentally different variants of snaking are investigated from a new point of view. This work is very interesting and worth of investigation. Only minor revisions are needed to make it ready for publication.

1. Thin-walled structures are generally sensitive to initial geometrical imperfections, which have a significant influence on the post-buckling path. In the proposed framework, how to qualify the stiffness reduction by imperfections is very challenging. Please add some further discussions on this issue.
2. Page 18, line 32, the range of K should be $0.01 < K < 0.02$, rather than $0.1 < K < 0.2$.
3. The reviewer is interested in how is the value of K ($K = 0.011405$) on page 18 determined? Was it derived from the formula or based on the finite element results?
4. In this manuscript, the "Maxwell tipping point" is an important conceptual definition, and the authors give a detailed definition of Maxwell tipping point in Sec. 4 (d). Could you please explain on the role of Maxwell tipping point in the instability of the shell or the physical significance of Maxwell tipping point in the instability of the shell? Because the authors just give the definition based on block of 4 elements shown in Fig. 13., but not based on the buckling behavior of the cylindrical shell.
5. Fig.1 (b) cannot display properly. Please check it.
6. The lines of ① ② ③ in Figure 8 should use different colors to highlight the results. The display of the energy hump display is not very intuitive.

Referee: 2

Comments to the Author(s)

The submission by the authors presents a new insight in the mechanics of the elastic initial post-buckling of an axially-loaded thin-walled cylindrical shell with the clever assistance of spring and rigid-link models. The insight concerns how the initial row of buckles is produced in a sequence of instabilities that is usually assumed to be triggered instantaneously. However, previously recorded (and indeed well-known) high-speed camera footage has revealed this to be a process involving a sequential buckling phenomenon along the circumference of the cylinder and the present study attributes this to be a "snaking-type", sometimes known as "cellular buckling-type", process. Recent generalized path-following finite element work by one of the co-authors has revealed the highly unstable equilibrium paths that the system contains, although since these are all unstable under rigid loading they are usually missed by physical instrumentation and naked eye observation. In the current work, the Maxwell stability criterion and a new concept, coined as the "Maxwell tipping point" is determined to predict whether a sequential buckling occurs or not.

The submission is very well presented, contains new and tangible mechanical insights, and is recommended for publication in Proceedings A with minor revisions, which are detailed below. (Note that line numbers refer to the numbers on the left hand side of the submitted manuscript).

Pages 3 (line 20) and 9 (line 16): Ohm's law for resistors is an often stated analogy for springs. However, a more exacting analogy is that of capacitors rather than resistors, since capacitors in parallel behave analogously to springs in parallel (effective stiffnesses and capacitances are obtained by summing those of individual elements), while capacitors in series behave analogously to springs in series (effective stiffnesses and capacitances are obtained by summing the reciprocal quantities of individual elements). In resistors, the series and parallel laws are the reverse of the spring stiffness laws. Suggest that this is amended.

Page 4 (line 43): The odd-integer sequence ending in "10" seems a little strange. Presumably "10" is the maximum number of dimples in the cylinder. Suggest that this is rephrased.

Page 5 - Figure 1 (please also see attached file showing marked up page 5):

- 1) In (a) can the black dots in the denoting the critical points be made slightly larger so that they are properly visible?
- 2) In (b), in the PDF that was received, the sequence of images appears to be broken.

Page 5, line 55: Replace "practise" with "practice".

Page 8 - Figure 4(b): There appears to be a misplaced label half-way up the vertical axis. Please check and remove if necessary.

Page 9, lines 7-8: Insert "to" between "(" and "heat, friction, etc.),"

Page 14, final paragraph above Section 4(b), which discusses stretching energies in cylindrical shell dimples was difficult to visualize. Could an additional diagram be included in Figure 11 to assist in visualizing the explanation?

Page 17 (line 31): Surely the range " $0.1 < K < 0.2$ " should be " $0.01 < K < 0.02$ "?

Page 19 (line 14 - Table 1 caption, and line 38): There appear to be an excessive number of closing parentheses in both these expressions.

Page 19 (line 49): Replace "to spontaneously transition" with "to transition spontaneously".

Page 20 (line 46): Replace "to further elucidate the hidden mechanics of this buckling process" with "to elucidate the hidden mechanics of this buckling process further".

Page 21 - Reference 13: Replace "Groh R. M. J." with "R. M. J. Groh", and "Ahmer M. Wadee" with "M. Ahmer Wadee".

Referee: 3

Comments to the Author(s)

This manuscript focuses on the instantaneous buckling procedure of a thin-walled cylindrical shell under axial compression, from pre-buckling to a single dimple and eventually to a complete ring of dimples along the circumferential direction, which is not easy to be captured experimentally and has drawn little attention. The authors started from the thin cylindrical shell under axial compression using finite element method and then analyzed the deformation mode transition. When instability occurs, a single dimple appears in the shell surface first. In the following process a sequence dimples formed successively. To explore the hidden mechanics, the authors turn to the spring systems coupled with adjacent cells. A two-dimensional cellular model, composed of snap-back arch cells connected in parallel and in series, was proposed to replicate the nonlinear response and illustrate the mechanism behind this phenomenon. The Maxwell displacement plays an important role in the mode selection, which denotes two stable states having equal energy. For some special case, the Maxwell displacement may overlap during multiple modes transformations, which is defined as Maxwell tipping point. This simple model

could rationally explain relevant experimental results. The topic of this paper is of interest, which provides a novel insight into the understanding of the complex instability behavior of shell.

Small things:

1. Generally, the Maxwell displacement has a close relationship with the energy of transition modes. It would be better to draw an energy landscape of spring system to provide an overall view and understanding.
2. It seems that Fig. 1(b) is incomplete.
3. Why the initial pre-buckling stiffness of tri-linear cells in series (Fig. 2) is set as infinite, while initial stiffness of cells in parallel (Fig. 4) is finite k_1 ?
4. It is mentioned that the cylinder can undergo transitions from a single dimple to higher-order dimples in two different ways: by traversing the energy barrier associated to a ladder (from odd to even) or an unstable snaking portion (from odd to odd). In experiments which path is observed and why? Can it be controlled experimentally?
5. Earlier literatures have suggested that the Batdorf parameter Z could affect the post-buckling behavior of cylindrical shells. For other smaller or larger Z , can the results be different?
6. "say for $uR/Lt = 0.34$ " in Page 6 may be incorrect according to Fig. 1(a).

Board Member:

Comments to Author(s):

Three review reports of the referenced manuscript were received. All of them are positive, and some minor comments/issues are provided. I agree with their judgement, and recommend publication after minor revisions.

Author's Response to Decision Letter for (RSPA-2020-0273.R0)

See Appendix A.

Decision letter (RSPA-2020-0273.R1)

24-Jun-2020

Dear Dr Dodwell

I am pleased to inform you that your manuscript entitled "Maxwell tipping points: the hidden mechanics of an axially compressed cylindrical shell" has been accepted in its final form for publication in Proceedings A.

Our Production Office will be in contact with you in due course. You can expect to receive a proof of your article soon. Please contact the office to let us know if you are likely to be away from e-mail in the near future. If you do not notify us and comments are not received within 5 days of sending the proof, we may publish the paper as it stands.

Open access

You are invited to opt for open access, our author pays publishing model. Payment of open access fees will enable your article to be made freely available via the Royal Society website as soon as it is ready for publication. For more information about open access please visit http://royalsocietypublishing.org/site/authors/open_access.xhtml. The open access fee for this journal is £1700/\$2380/€2040 per article. VAT will be charged where applicable.

Note that if you have opted for open access then payment will be required before the article is published – payment instructions will follow shortly.

If you wish to opt for open access then please inform the editorial office (proceedingsa@royalsociety.org) as soon as possible.

Your article has been estimated as being 22 pages long. Our Production Office will inform you of the exact length at the proof stage.

Proceedings A levies charges for articles which exceed 20 printed pages. (based upon approximately 540 words or 2 figures per page). Articles exceeding this limit will incur page charges of £150 per page or part page, plus VAT (where applicable).

Under the terms of our licence to publish you may post the author generated postprint (ie. your accepted version not the final typeset version) of your manuscript at any time and this can be made freely available. Postprints can be deposited on a personal or institutional website, or a recognised server/repository. Please note however, that the reporting of postprints is subject to a media embargo, and that the status the manuscript should be made clear. Upon publication of the definitive version on the publisher's site, full details and a link should be added.

You can cite the article in advance of publication using its DOI. The DOI will take the form: 10.1098/rspa.XXXX.YYYY, where XXXX and YYYY are the last 8 digits of your manuscript number (eg. if your manuscript number is RSPA-2017-1234 the DOI would be 10.1098/rspa.2017.1234).

For tips on promoting your accepted paper see our blog post:
<https://blogs.royalsociety.org/publishing/promoting-your-latest-paper-and-tracking-your-results/>

On behalf of the Editor of Proceedings A, we look forward to your continued contributions to the Journal.

Sincerely,
Raminder Shergill
proceedingsa@royalsociety.org

Appendix A

Revisions for ‘Maxwell tipping points: the hidden mechanics of an axially compressed cylindrical shell’

G. W. Hunt, R. M. J. Groh & T. J. Dodwell

May 27, 2020

We thank the reviewers and editors for their supportive comments, and minor suggestions for the revision. Each has been addressed. A comment-by-comment summary of our edits is provided below.

Referee 1

1. Thin-walled structures are generally sensitive to initial geometrical imperfections, which have a significant influence on the post-buckling path. In the proposed framework, how to qualify the stiffness reduction by imperfections is very challenging. Please add some further discussions on this issue.
Thank you for your suggestion. A short discussion has been added in the introduction.
2. Page 18, line 32, the range of K should be $0.01 < K < 0.02$, rather than $0.1 < K < 0.2$.
Amended as suggested.
3. The reviewer is interested in how is the value of K ($K = 0.011405$) on page 18 determined? Was it derived from the formula or based on the finite element results?
This value was determined by changing K by trial and error. This has been clarified and added to the text.
4. In this manuscript, the Maxwell tipping point is an important conceptual definition, and the authors give a detailed definition of Maxwell tipping point in Sec. 4 (d). Could you please explain on the role of Maxwell tipping point in the instability of the shell or the physical significance of Maxwell tipping point in the instability of the shell? Because the authors just give the definition based on block of 4 elements shown in Fig. 13., but not based on the buckling behavior of the cylindrical shell.
Thank you for the suggestion. We have added the following discussion in Section 5. “In summary, our analysis shows that the buckling behaviour of Yamaki’s longest cylinder ($Z = 1000$) [16] lies beyond the Maxwell tipping point. This means the cylinder is most likely to stabilise into a fully formed ring of diamonds upon buckling, and this is indeed what is observed in Yamaki’s experiments. For other cylinder geometries, i.e. Batdorf parameter Z , the buckling behaviour may precede the Maxwell tipping point. Although such a parametric study is not investigated herein, Yamaki’s experiments on other geometries $Z = [50, 100, 200, 500]$ suggest similar behaviour to the case of $Z = 1000$ studied herein [16,p. 236]”
5. Fig.1 (b) cannot display properly. Please check it.
For some reason the PDF did not render the figure correctly. This has been amended in the revision.
6. The lines of (1), (2) and (3) in Figure 8 should use different colors to highlight the results. The display of the energy hump display is not very intuitive.
The lines are now rendered in different colours.

Referee 2

1. Pages 3 (line 20) and 9 (line 16): Ohm’s law for resistors is an often stated analogy for springs. However, a more exacting analogy is that of capacitors rather than resistors, since capacitors in parallel behave analogously to springs in parallel (effective stiffnesses and capacitances are obtained by summing those of individual elements), while capacitors in series behave analogously to springs in series (effective stiffnesses and capacitances are obtained by summing the reciprocal quantities of individual elements). In resistors, the series and parallel laws are the reverse of the spring stiffness laws. Suggest that this is amended.
We have amended this in the text as suggested by referring to both resistors and capacitors.
2. Page 4 (line 43): The odd-integer sequence ending in ”10” seems a little strange. Presumably ”10” is the maximum number of dimples in the cylinder. Suggest that this is rephrased.
This has been amended as suggested.

3. Page 5 - Figure 1 (please also see attached file showing marked up page 5):
 - (a) In (a) can the black dots in the denoting the critical points be made slightly larger so that they are properly visible?
 - (b) In (b), in the PDF that was received, the sequence of images appears to be broken.

Both amended as suggested.
4. Page 5, line 55: Replace "practise" with "practice".
Amended as suggested.
5. Page 8 - Figure 4(b): There appears to be a misplaced label half-way up the vertical axis. Please check and remove if necessary.
Amended as suggested.
6. Page 9, lines 7-8: Insert "to" between "(" and "heat, friction, etc.),"
Amended as suggested.
7. Page 14, final paragraph above Section 4(b), which discusses stretching energies in cylindrical shell dimples was difficult to visualize. Could an additional diagram be included in Figure 11 to assist in visualizing the explanation?
Thank you for the comment. We thought about different ways of adding a diagram to elucidate our argument more clearly, however, we have decided against this for two reasons. First, the cylinder is a high-dimensional problem and in our case discretised by nodal degrees of freedom, rather than amplitudes of modes. Therefore a diagrammatic illustration of stretching and bending energies corresponding to modes would be difficult to construct. Second, we are not convinced that such a diagram would necessarily provide much insight, and rather, think it would be more misleading than helpful.
8. Page 17 (line 31): Surely the range " $0.1 < K < 0.2$ " should be " $0.01 < K < 0.02$ "?
Amended as suggested.
9. Page 19 (line 14 - Table 1 caption, and line 38): There appear to be an excessive number of closing parentheses in both these expressions.
Amended as suggested.
10. Page 19 (line 49): Replace "to spontaneously transition" with "to transition spontaneously".
Amended as suggested.
11. Page 20 (line 46): Replace "to further elucidate the hidden mechanics of this buckling process" with "to elucidate the hidden mechanics of this buckling process further".
Amended as suggested.
12. Page 21 - Reference 13: Replace 'Groh R. M. J.' with 'R. M. J. Groh', and 'Ahmer M. Wadee' with 'M. Ahmer Wadee'.
Amended as suggested.

Referee 3

1. Generally, the Maxwell displacement has a close relationship with the energy of transition modes. It would be better to draw an energy landscape of spring system to provide an overall view and understanding.
Amended as suggested in figure 6.
2. It seems that Fig. 1(b) is incomplete.
Amended as suggested.
3. Why the initial pre-buckling stiffness of tri-linear cells in series (Fig. 2) is set as infinite, while initial stiffness of cells in parallel (Fig. 4) is finite k_1 ? Finite pre-buckling stiffness is necessary for the snapback effect. The text has been amended to explain this point.
4. It is mentioned that the cylinder can undergo transitions from a single dimple to higher-order dimples in two different ways: by traversing the energy barrier associated to a ladder (from odd to even) or an unstable snaking portion (from odd to odd). In experiments which path is observed and why? Can it be controlled experimentally?
Thank your for raising this point. It is more likely that the energy barrier associated to the ladder is traversed as it represents a smaller energy barrier (one additional dimple rather than two), as stated in the second paragraph of Section 5. This could be tested experimentally by observing the dynamics in high-speed photography experiments in detail, but without many additional actuators that control the buckling sequence quasi-statically, this would be difficult to control directly. No further changes have been made in the text.

5. Earlier literatures have suggested that the Batdorf parameter Z could affect the post-buckling behavior of cylindrical shells. For other smaller or larger Z , can the results be different?
Thank you for the comment. This was also raised by referee 1. Please see our response to comment 4 of referee 1 above.
6. ‘say for $uR/Lt = 0.34$ ’ in Page 6 may be incorrect according to Fig. 1(a).
In the sentence referred to, we are discussing a transition from the stable pre-buckling state to the stable single dimple via the unstable single dimple. This transition is possible for $uR/Lt = 0.34$, and indeed over a wide region of uR/Lt , as shown in figure 1(a). No further changes have been made in the text.